# TOPOLOGY-AWARE ROBUST OPTIMIZATION FOR OUT-OF-DISTRIBUTION GENERALIZATION

**Fengchun Qiao**
University of Delaware
`fengchun@udel.edu`

**Xi Peng**
University of Delaware
`xipeng@udel.edu`

## ABSTRACT

Out-of-distribution (OOD) generalization is a challenging machine learning problem yet highly desirable in many high-stake applications. Existing methods suffer from overly pessimistic modeling with low generalization confidence. As generalizing to arbitrary test distributions is impossible, we hypothesize that further structure on the topology of distributions is crucial in developing strong OOD resilience. To this end, we propose topology-aware robust optimization (TRO) that seamlessly integrates distributional topology in a principled optimization framework. More specifically, TRO solves two optimization objectives: (1) Topology Learning which explores data manifold to uncover the distributional topology; (2) Learning on Topology which exploits the topology to constrain robust optimization for tightly-bounded generalization risks. We theoretically demonstrate the effectiveness of our approach, and empirically show that it significantly outperforms the state of the arts in a wide range of tasks including classification, regression, and semantic segmentation. Moreover, we empirically find the data-driven distributional topology is consistent with domain knowledge, enhancing the explainability of our approach.

## 1 INTRODUCTION

Recent years have witnessed a surge of applying machine learning (ML) in high-stake and safety-critical applications. Such applications pose an unprecedented *out-of-distribution (OOD) generalization challenge*: ML models are constantly exposed to unseen distributions that lie outside their training space. Despite well-documented success for *interpolation*, modern ML models (*e.g.*, deep neural networks) are notoriously weak for *extrapolation*; a highly accurate model on average can fail catastrophically when presented with rare or unseen distributions (Arjovsky et al., 2019). For example, a flood predictor, trained with data of all 89 major flood events in the U.S. from 2000 to 2020, would erroneously predict on event "Hurricane Ida" in 2021. Without addressing this challenge, it is unclear when and where a model can be applied and how much risk is associated with its use.

A promising solution for out-of-distribution generalization is to conduct *distributionally robust optimization* (DRO) (Namkoong & Duchi, 2016; Staib & Jegelka, 2019; Levy et al., 2020). DRO minimizes the *worst-case* expected risk over an *uncertainty set* of potential test distributions. The uncertainty set is typically formulated as a divergence ball surrounding the training distribution endowed with a certain distance metric such as $f$-divergence (Namkoong & Duchi, 2016) and Wasserstein distance (Shafieezadeh Abadeh et al., 2018). Compared to empirical risk minimization (ERM) (Vapnik, 1998) that minimizes the average loss, DRO is more robust against *distributional drifts* from spurious correlations, adversarial attacks, subpopulations, or naturally-occurring variation (Robey et al., 2021).

However, it is non-trivial to build a realistic uncertainty set that truly approximates unseen distributions. On the one hand, to confer robustness against extensive distributional drifts, the uncertainty set has to be sufficiently large, which increases the risks of conferring implausible distributions, *e.g.*, outliers, and thus yielding overly pessimistic models with low prediction confidence (Hu et al., 2018; Frogner et al., 2021). On the other hand, the worst-case distributions are not necessarily the *influential* ones that are truly connected to unseen distributions; optimizing over worst-case rather than influential distributions would yield compromised OOD resilience.

---

[1]The source code and pre-trained models are available at: `https://github.com/joffery/TRO`.

As generalizing to arbitrary test distributions is impossible, we hypothesize further structure on the topology of distributions is crucial in constructing a realistic uncertainty set. More specifically, we propose topology-aware robust optimization (TRO) by integrating two optimization objectives:

(1) **Topology learning**: We model the data distributions as many discrete groups lying on a common low-dimensional manifold, where we can *explore* the distributional topology by either using physical priors or measuring multiscale Earth Mover's Distance (EMD) among distributions.

(2) **Learning on topology**: The acquired distributional topology is then *exploited* to construct a realistic uncertainty set, where robust optimization is constrained to bound the generalization risk within a topology graph, rather than blindly generalizing to unseen distributions.

Our contributions include: 1. A new principled optimization method that seamlessly integrates topological information to develop strong OOD resilience. 2. Theoretical analysis that proves our method enjoys fast convergence for both convex and non-convex loss functions while the generalization risk is tightly bounded. 3. Empirical results in a wide range of tasks including classification, regression, and semantic segmentation that demonstrate the superior performance of our method over SOTA. 4. Data-driven distributional topology that is consistent with domain knowledge and facilitates the explainability of our approach.

## 2 PROBLEM FORMULATION AND PRELIMINARY WORKS

The problem of out-of-distribution (OOD) generalization is defined by a pair of random variables $(X, Y)$ over instances $x \in \mathcal{X} \subseteq \mathbb{R}^d$ and corresponding labels $y \in \mathcal{Y}$, following an unknown joint probability distribution $P(X, Y)$. The objective is to learn a predictor $f \in \mathcal{F}$ such that $f(x) \to y$ for any $(x, y) \sim P(X, Y)$. Here $\mathcal{F}$ is a function class that is model-agnostic for a prediction task. However, unlike typical supervised learning, the OOD generalization is complicated since one cannot sample directly from $P(X, Y)$. Instead, it is assumed that we can only measure $(X, Y)$ under different environmental conditions $e$ so that data is drawn from a set of groups $\mathcal{E}_{\text{all}}$ such that $(x, y) \sim P_e(X, Y)$. For example, in flood prediction, these environmental conditions denote the latent factors (*e.g.*, stressors, precipitation, terrain, etc) that underlie different flood events. Let $\mathcal{E}_{\text{train}} \subsetneq \mathcal{E}_{\text{all}}$ be a finite subset of training groups (distributions), given the loss function $\ell$, an OOD-resilient model $f$ can be learned by solving a minimax optimization:

$$\min_{f \in \mathcal{F}} \left\{ \mathcal{R}(f) := \sup_{e \in \mathcal{E}_{\text{all}}} \mathbb{E}_{(x,y) \sim P_e(X,Y)}[\ell(f(x), y)] \right\}. \tag{1}$$

Intuitively, Eq. 1 aims to learn a model that minimizes the worst-case risk over the entire family $\mathcal{E}_{\text{all}}$. It is nontrivial since we do not have access to data from any unseen distributions $\mathcal{E}_{\text{test}} = \mathcal{E}_{\text{all}} \backslash \mathcal{E}_{\text{train}}$.

**Empirical Risk Minimization** (ERM). Typically, classic supervised learning employs ERM (Vapnik, 1998) to find a model $f$ that minimizes the *average* risk under the training distribution $P_{tr}$:

$$\min_{f \in \mathcal{F}} \{ \mathcal{R}(f) := \mathbb{E}_{(x,y) \sim P_{tr}}[\ell(f(x), y)] \}.$$

Though proved to be effective in *i.i.d.* settings, models trained via ERM heavily rely on spurious correlations that do not always hold under distributional drifts (Arjovsky et al., 2019).

**Distributionally Robust Optimization** (DRO). To develop OOD resilience, DRO (Namkoong & Duchi, 2016) minimizes the *worst-case* risk over an uncertainty set $Q$ by solving:

$$\min_{f \in \mathcal{F}} \{ \mathcal{R}(f) := \sup_{Q \in \mathcal{P}(P_{tr})} \mathbb{E}_{(x,y) \sim Q}[\ell(f(x), y)] \}. \tag{2}$$

Here the uncertainty set $Q$ approximates potential test distributions. It is usually formulated as a divergence ball with a radius of $\rho$ surrounding the training distribution $\mathcal{P}(P_{tr}) = \{Q : D(Q, P_{tr}) \leq \rho\}$ endowed with a certain distance metric $D(\cdot, \cdot)$ such as $f$-divergence (Namkoong & Duchi, 2016) or Wasserstein distance (Shafieezadeh Abadeh et al., 2018). To construct a realistic uncertainty set without being overly conservative, Group DRO is further developed to formulate the uncertainty set as the mixture of training groups (Hu et al., 2018; Sagawa et al., 2019).

Despite the well-documented success, existing DRO methods suffer from critical limitations. (1) To endow robustness against a wide range of potential test distributions, the radius of the divergence ball has to be sufficiently large with high risks of containing implausible distributions; optimizing

for implausible distributions would fundamentally damage the OOD resilience by yielding overly-pessimistic models with low prediction confidence. (2) The worst-case groups are not necessarily the *influential* ones that are truly connected to unseen distributions; optimizing over worst-case rather than influential groups would yield compromised OOD resilience.

## 3 TOPOLOGY-AWARE ROBUST OPTIMIZATION

We propose a new principled optimization method (TRO) to develop OOD resilience, which integrates topology and optimization via a two-phase scheme: *Topology Learning* and *Learning on Topology*.

### 3.1 TOPOLOGY LEARNING: EXPLORE THE DISTRIBUTIONAL TOPOLOGY

We model the data groups $\mathcal{E}_{\text{all}}$ as many discrete distributions lying on a common low-dimensional manifold in a high-dimensional data measurement space. In such case their structure, *i.e. distributional topology*, can be naturally captured by a graph $\mathcal{G} = (V, E)$, where the entities $V = \cup_{e \in \mathcal{E}_{\text{all}}} X_e$ symbolize the groups and the edges $E$ represent interactions among groups. The topology graph is constructed by: (1) *identifying entity*: we assume the entities are defined by the given group identities; and (2) *uncovering interactions*: we consider two scenarios to measure the connectivity between discrete distributions as illustrated in Fig 1.

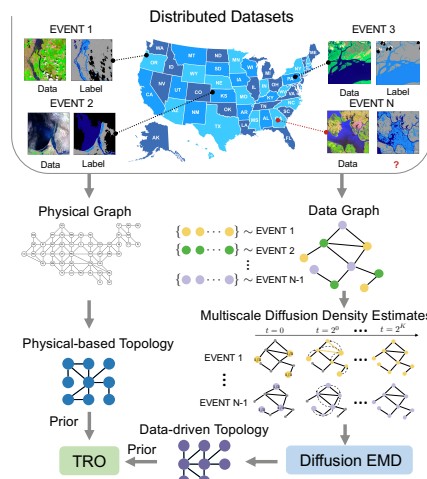

**Physical-based distributional topology**. In the scenario where the distributional adjacency information is available, we can instantly acquire the topology $\mathcal{G}_{physic}$ by simply imposing the predefined neighborhood information. For example, to capture the similarity of weather events in the U.S., one can construct a graph where each state realizes an entity, and the physical adjacency be-

Figure 1: Overview of topology-aware distributionally robust optimization (TRO).

tween two states results in an edge (see Fig. 1). In this case, $\mathcal{G}_{physic}$ functions as *a physical prior* to constrain the robust optimization introduced in Sec. 3.2. We empirically find $\mathcal{G}_{physic}$ yields an improvement of 9.56% over the state of the art regarding OOD generalization reported in Sec. 5.1.

**Data-driven distributional topology**. In the absence of $\mathcal{G}_{physic}$, we propose a data-driven approach to learn the topology $\mathcal{G}_{data}$ from training data. Specifically, we embed the individual groups onto a shared data graph based on an affinity matrix of the combined data. Inspired by Leeb & Coifman (2016), such a data graph can be viewed as a discretization of an underlying Riemann closed manifold. By simulating a time-dependent diffusion process over the graph, we will obtain density estimates at multiple scales for each group, which will be used to calculate $\ell_1$ distances between two groups. Such multiscale $\ell_1$ distance has been proved to be topologically equivalent to the Earth Mover's Distance (EMD) on the manifold geodesic, but cutting down the computational complexity from $O\left(m^2 n^3\right)$ to $\tilde{O}(mn)$ between $m$ distributions over $n$ data points (Tong et al., 2021).

We obtain the data-driven topology through three steps: (1) *Data graph construction*: we construct a data graph through an affinity matrix $\mathbf{K}$ of the combined data. $\mathbf{K}$ can be implemented through kernel functions (*e.g.*, RBF kernel) which capture the similarity of data. Instead of calculating the similarity between raw data, we calculate the similarity between features extracted from an ERM-trained model as it captures spurious correlations which preserve group identity (Creager et al., 2021). Specifically, we define the affinity matrix as: $\mathbf{K}_{i,j} = \exp\left(-\|f(x_i) - f(x_j)\|^2/\sigma^2\right)$, where $\sigma^2$ is the kernel scale. (2) *Multiscale diffusion density estimation*: to simulate the diffusion process over the graph, we obtain a Markov diffusion operator $\mathbf{P}$ from $\mathbf{K}$. Following Coifman & Lafon (2006), we normalize the affinity matrix: $\mathbf{M} = \mathbf{Q}^{-1}\mathbf{K}\mathbf{Q}^{-1}$, where $\mathbf{Q}$ is a diagonal matrix and $\mathbf{Q}_{i,i} = \sum_j \mathbf{K}_{i,j}$. The diffusion operator is defined as $\mathbf{P} = \mathbf{D}^{-1}\mathbf{M}$, where $\mathbf{D}$ is a diagonal matrix and $\mathbf{D}_{i,i} = \sum_j \mathbf{M}_{i,j}$. The operator $\mathbf{P}$ will be used to approximate the multiscale density estimates $\mu_e$ for each data group $X_e$: $\mu_e^t = \frac{1}{n_e}\mathbf{P}^t\mathbf{1}_{X_e}$, where $t$ is the diffusion time, $\mathbf{P}^t$ denotes the $t$-th power of $\mathbf{P}$, and $\mathbf{1}_{X_e}$ is the

indicator function for group $e$. Intuitively, $\mathbf{P}^t_{i,j}$ sums the probabilities of all possible paths of length $t$ between $x_i$ and $x_j$. By taking multiple powers of $\mathbf{P}$, $\mu_e$ reveals the topological structure of $X_e$ at multiple scales. (3) *Diffusion EMD measurement*: we follow Tong et al. (2021) to measure the geodesic distance $W_{\alpha,K}(X_e, X_{e'})$ between $X_e$ and $X_{e'}$ by aggregating the $\ell_1$ distances between the multiscale density estimates:

$$W_{\alpha,K}(X_e, X_{e'}) = \sum_{k=0}^{K} \|T_{\alpha,k}(X_e) - T_{\alpha,k}(X_{e'})\|_1, \tag{3}$$

where $\alpha$ is used to balance long- and short-range distances, $K$ is the maximum scale, and

$$T_{\alpha,k}(X_e) = \begin{cases} 2^{-(K-k-1)\alpha}\left(\mu_e^{\left(2^{k+1}\right)} - \mu_e^{\left(2^k\right)}\right), & k < K \\ \mu_e^{\left(2^K\right)}, & k = K \end{cases}$$

Although $\mathcal{G}_{data}$ is computationally more expensive than $\mathcal{G}_{physic}$, our experimental results in Sec. 5.2 indicate that optimizing with $\mathcal{G}_{data}$ can yield improved OOD resilience. Besides, the ablation study in Sec. 5.3 also indicates that $\mathcal{G}_{data}$ is consistent with domain knowledge and enhances the explainability of TRO. Last but not least, the data-driven method is fully differentiable, making it amenable to jointly conducting topology learning and learning on topology in an end-to-end manner. We leave this as future work.

## 3.2 LEARNING ON TOPOLOGY: EXPLOIT TOPOLOGY FOR ROBUST OPTIMIZATION

Next, we propose a principled method that integrates distributional topology to develop TRO. The key challenge is how to leverage $\mathcal{G}$ to construct a *uncertainty set* which can approximate unseen distributions with bounded generalization risk. Our main idea is to assess the *group centrality* of training distributions. *Graph centrality* is widely used in social network analysis (Newman, 2005) to measure how much information is propagated through each entity. Here we introduce *group centrality* to identify *influential groups* that are truly connected to unseen distributions, which can be calculated using graph measurements (Tian et al., 2019) such as degree, betweenness, and closeness. More specifically, we first calculate the centrality of each entity in $\mathcal{G}$ as a *topological prior* $\mathbf{p}$ to identify influential groups. Then, we construct the *uncertainty set* as an arbitrary mixture of training groups $Q := \{\sum_{e \in \mathcal{E}_{train}} q_e P_e \mid \mathbf{q} \in \Delta_m\}$ where $q_e$ denotes the weight of group $e$, $P_e$ is the distribution of group $e$, and $\Delta_m$ is a $(m-1)-$dimensional probability simplex. Finally, we use the prior $\mathbf{p}$ to constrain the uncertainty set $Q$ by solving the minimax optimization problem as:

---

**Algorithm 1:** TRO Algorithm

**Input:** Data of $\mathcal{E}_{train}$, Step sizes $\eta_\theta$ and $\eta_{\mathbf{q}}$
**Output:** Learned model $f$
*Topology Learning*:
**if** $\mathcal{G}_{physic}$ *exists* **then**
   | $\mathcal{G} \leftarrow \mathcal{G}_{physic}$
**else**
   | Obtain the affinity matrix $\mathbf{K}$ from data
   | $\mathbf{Q} \leftarrow \text{Diag}\left(\sum_j \mathbf{K}_{ij}\right)$
   | $\mathbf{M} \leftarrow \mathbf{Q}^{-1}\mathbf{K}\mathbf{Q}^{-1}$
   | $\mathbf{D} \leftarrow \text{Diag}\left(\sum_j \mathbf{M}_{ij}\right)$
   | $\mathbf{P} \leftarrow \mathbf{D}^{-1}\mathbf{M}$
   | Obtain $\mathcal{G}_{data}$ via Eq. 3
   | $\mathcal{G} \leftarrow \mathcal{G}_{data}$
**end**
*Learning on Topology*:
Calculate topological prior $\mathbf{p}$ from $\mathcal{G}$
**while** *not converged* **do**
   | Sample $(x, y) \sim P_e(X, Y) \, \forall e \in \mathcal{E}_{train}$
   | Calculate $\mathcal{R}(f, \mathbf{q})$ via Eq. 5
   | Update $\theta$ and $\mathbf{q}$ via Eq. 6
**end**

---

$$\min_{f \in \mathcal{F}}\{\mathcal{R}(f, \mathbf{q}) := \max_{\mathbf{q} \in \Delta_m} \sum_{e \in \mathcal{E}_{train}} q_e \, \mathbb{E}_{(x,y) \sim P_e(X,Y)}[\ell(f(x), y)]\}, \quad \text{s.t. } \mathcal{D}(\mathbf{q}\|\mathbf{p}) \leq \tau. \tag{4}$$

Intuitively, groups with high training loss and centrality will be assigned with large weights; this can tightly bound the OOD generalization risk within a topological graph. $\mathcal{D}$ is an arbitrary distributional distance metric. We use $\ell_2$ distance to implement $\mathcal{D}$ due to its strong convexity and simplicity.

However, solving Eq. 4 often leads to a non-convex problem, wherein methods such as stochastic gradient descent (SGD) cannot guarantee constraint satisfaction (Robey et al., 2021). To address this

issue, we leverage Karush–Kuhn–Tucker conditions (Boyd et al., 2004) and introduce a Lagrange multiplier to convert the constrained problem into its unconstrained counterpart:

$$\min_{f \in \mathcal{F}} \{ \mathcal{R}(f, \mathbf{q}) := \max_{\mathbf{q} \in \Delta_m} \sum_{e \in \mathcal{E}_{\text{train}}} q_e \, \mathbb{E}_{(x,y) \sim P_e(X,Y)} [\ell(f(x), y)] - \lambda \mathcal{D}(\mathbf{q} \| \mathbf{p}) \}, \quad (5)$$

where $\lambda$ is the dual variable. Let $\theta \in \Theta$ be the model parameters of $f$, we can solve the primal-dual problem effectively by alternatively updating:

$$\theta^{t+1} = \theta^t - \eta_\theta^t \nabla_\theta \mathcal{R}(f, \mathbf{q}), \quad \mathbf{q}^{t+1} = \mathcal{P}_{\Delta_m}(\mathbf{q}^t + \eta_{\mathbf{q}}^t \nabla_{\mathbf{q}} \mathcal{R}(f, \mathbf{q})), \quad (6)$$

where $\eta_\theta^t$ ($\eta_{\mathbf{q}}^t$) is gradient descent (ascent) step size. $\mathcal{P}_{\Delta_m}(\mathbf{q})$ projects $\mathbf{q}$ onto simplex $\Delta_m$ for regularization. The overall algorithm of TRO is shown in Alg. 1. In Sec. 4, we show TRO enjoys fast convergence for both convex and non-convex loss functions, while the generalization risk is tightly bounded with topological constraints. We empirically demonstrate TRO achieves strong OOD resilience by striking a good balance between the worst-case and influential groups (see Sec. 5.2).

**Calculation of group centrality**. We use betweenness centrality to measure the centrality of groups. Betweenness centrality measures how often an entity is on the shortest path between two other entities in the topology. Freeman (1977) reveals that entities with higher betweenness centrality would have more control over the topology as more information will pass through them. For physical-based topology $\mathcal{G}_{physic}$, we define the centrality of group $e$ by computing the fraction of shortest paths that pass through it: $c_e^{physic} = \sum_{s \in \mathcal{E}_{\text{train}}, t \in \mathcal{E}_{\text{test}}} \frac{\sigma(s,t|e)}{\sigma(s,t)}$, where $\sigma(s,t)$ is the number of shortest paths between groups $s$ and $t$ in the graph ($(s,t)$-paths), and $\sigma(s, t \mid e)$ is the number of $(s,t)$-paths that go through group $e$. Intuitively, $c_e^{physic}$ measures how much information is propagated through $e$ from the start (training) to the end (test). For data-driven topology $\mathcal{G}_{data}$, the underlying assumption is that training groups with high centrality also exert strong influence on unseen groups. Instead of sampling group pairs from two separate sets, we sample $(s,t)$ from $\mathcal{E}_{\text{train}}$. The centrality is modified as: $c_e^{data} = \sum_{s,t \in \mathcal{E}_{\text{train}}} \frac{\sigma(s,t|e)}{\sigma(s,t)}$. We use softmax function to normalize $c_e$ and the prior probability for group $e \in \mathcal{E}_{\text{train}}$ is defined as: $p_e = \exp(c_e) / \sum_{e \in \mathcal{E}_{\text{train}}} \exp(c_e)$.

# 4 THEORETICAL ANALYSIS

## 4.1 CONVERGENCE ANALYSIS

In this section, we show that by choosing appropriate step sizes $\eta_\theta^t$ and $\eta_q^t$, TRO yields fast convergence rates for both convex and non-convex loss functions. We first state the assumptions of the theorems. Next, we give the convergence rate for convex loss functions in Theorem 1 and the convergence rate for non-convex loss functions in Theorem 2.

**Definition 1.** (Lipschitz continuity) A mapping $f : \mathcal{X} \to \mathbb{R}^m$ is $G$-Lipschitz continuous if for any $x, y \in \mathcal{X}, \|f(x) - f(y)\| \leq G\|x - y\|$.
**Definition 2.** (Smoothness) A function $f : \mathcal{X} \to \mathbb{R}$ is $L$-smooth if it is differentiable on $\mathcal{X}$ and the gradient $\nabla f$ is L-Lipschitz continuous, *i.e.*, $\|\nabla f(x) - \nabla f(y)\| \leq L\|x - y\|$ for all $x, y \in \mathcal{X}$.
**Assumption 1.** We make the following assumptions throughout the paper: Given $\theta$, the loss function $\ell(f_\theta(x), y)$ is G-Lipschitz continuous and L-smooth with respect to $x$.

**Convex Loss.** The expected number of stochastic gradient computations is utilized to estimate the convergence rate. To reach a duality gap of $\epsilon$ (Nemirovski et al., 2009), the optimal rate of convergence for solving the stochastic min-max problems is $\mathcal{O}\left(1/\epsilon^2\right)$ if it is convex-concave. The duality gap of the pair $(\tilde{\theta}, \tilde{q})$ is defined as $\max_{q \in \Delta_m} \mathcal{R}(\tilde{\theta}, q) - \min_{\theta \in \Theta} \mathcal{R}(\theta, \tilde{q})$. In the case of strong duality, $(\tilde{\theta}, \tilde{q})$ is optimal *iif* the duality gap is zero. We show TRO achieves the optimal $\mathcal{O}\left(1/\epsilon^2\right)$ rate.
**Theorem 1.** Consider the dual problem in Eq. 5 when the loss function is convex and Assumption 1 holds. Let $\Theta$ bounded by $R_\Theta$, $\mathbb{E}\left[\|\nabla_\theta \mathcal{R}(\theta, q)\|_2^2\right] \leq \hat{G}_\theta^2$, and $\mathbb{E}\left[\|\nabla_q \mathcal{R}(\theta, q)\|_2^2\right] \leq \hat{G}_q^2$. With step sizes $\eta_\theta = 2R_\Theta / \left(\hat{G}_\theta \sqrt{T}\right)$ and $\eta_q = 2/\left(\hat{G}_q \sqrt{T}\right)$, the output of TRO satisfies:

$$\mathbb{E}\left[\max_{q \in \Delta_m} \mathcal{R}\left(\theta_T, q\right) - \min_{\theta \in \Theta} \mathcal{R}\left(\theta, q_T\right)\right] \leq \frac{3R_\Theta \hat{G}_\theta + 3\hat{G}_q}{\sqrt{T}}. \quad (7)$$

Theorem 1 shows that our method requires $T = \mathcal{O}\left(1/\epsilon^2\right)$ iterations to reach a duality gap within $\epsilon$.

To derive the convergence rate for non-convex functions., we define $\epsilon$-stationary points as follows:

**Definition 3.** ($\epsilon$-stationary point) For a differentiable function $f : \mathcal{X} \rightarrow \mathbb{R}$, a point $x \in \mathcal{X}$ is said to be first-order $\epsilon$-stationary if $\|\nabla f(x)\| \leq \epsilon$.

**Nonconvex Loss.** The loss function $\ell(f_\theta(x), y)$ can be nonconvex and as a result, $\mathcal{R}(\theta, q)$ is nonconvex in $\theta$. Following Collins et al. (2020), we define $(\tilde{\theta}, \tilde{q})$ is an $(\epsilon, \delta)$-stationary point of $\mathcal{R}$ if: $\left\|\nabla_\theta \mathcal{R}(\tilde{\theta}, \tilde{q})\right\|_2 \leq \epsilon$ and $\mathcal{R}(\tilde{\theta}, \tilde{q}) \geq \max_{q \in \Delta_m} \mathcal{R}(\tilde{\theta}, q) - \delta$, where $\epsilon, \delta > 0$.

**Theorem 2.** If Assumption 1 holds and the loss function is bounded by $B$ and is M-smooth, the output of Alg. 1 satisfies:

$$\mathbb{E}\left[\|\nabla_\theta \mathcal{R}\left(\theta_T, q_T\right)\|_2^2\right] \leq \frac{\mathcal{R}\left(\theta^0, q^0\right) + B}{T\eta_\theta} + \frac{2\eta_q\sqrt{n}B\hat{G}_q}{\eta_\theta} + \frac{\eta_\theta M \hat{G}_\theta^2}{2},$$

$$\mathbb{E}\left[\mathcal{R}\left(\theta_T, q_T\right)\right] \geq \max_{q \in \Delta_m}\left\{\mathbb{E}\left[\mathcal{R}\left(\theta_T, q\right)\right]\right\} - \frac{1}{\eta_q T} - \frac{\eta_q \hat{G}_q^2}{2}.$$

(8)

Theorem 2 shows that our method converges in expectation to an $(\epsilon, \delta)$-stationary point of $\mathcal{R}$ in $\mathcal{O}(1/\epsilon^4)$ stochastic gradient evaluations.

## 4.2 GENERALIZATION BOUNDS

In this section, we provide learning guarantees for TRO. Compared to DRO, TRO achieves a lower upper bound on the generalization risks over unseen distributions with the topological constraint.

Let $\mathcal{H}$ denote the family of losses associated with a hypothesis set $\mathcal{F}$: $\mathcal{H} = \{(x, y) \mapsto \ell(f(x), y) : f \in \mathcal{F}\}$, and $\mathbf{n} = (n_1, \ldots, n_m)$ denote the vector of sample sizes for all training groups. Following Mohri et al. (2019), we define weighted Rademacher complexity for any $\mathcal{F}$ as:

$$\mathfrak{R}_{\mathbf{n}}(\mathcal{H}, q) = \mathop{\mathbb{E}}_{S_e \sim P_e} \mathop{\mathbb{E}}_{\boldsymbol{\sigma}}\left[\sup_{f \in \mathcal{F}} \sum_{e=1}^m \frac{q_e}{n_e} \sum_{i=1}^{n_e} \sigma_{e,i} \ell\left(f\left(x_{e,i}\right), y_{e,i}\right)\right],$$

where $e$ denotes group index, $S_e$ a sample of size $n_e$, $P_e$ the distribution of group $e$, and $\boldsymbol{\sigma} = (\sigma_{e,i})_{e \in [m], i \in [n_e]}$ a collection of Rademacher variables. The minimax weighted Rademacher complexity for a subset $\Lambda \subseteq \Delta_m$ is defined by $\mathfrak{R}_{\mathbf{n}}(\mathcal{H}, \Lambda) = \max_{q \in \Lambda} \mathfrak{R}_n(\mathcal{H}, q)$ where $n = \sum_{e=1}^m n_e$. Let $P_\Lambda$ be the distribution over the mixture of training groups and $\hat{P}_\Lambda$ be its empirical counterpart. Let the distribution of some test group be $P$. The learning guarantee for $P$ is shown in Theorem 3.

**Theorem 3.** Assume the loss function $\ell$ is bounded by $B$. For any $\epsilon \geq 0$ and $\delta > 0$, with probability at least $1 - \delta$, the following inequality holds for all $f \in \mathcal{F}$:

$$\mathcal{R}_P(f) \leq \mathcal{R}_{\hat{P}_\Lambda}(f, q) + 2\mathfrak{R}_{\mathbf{n}}(\mathcal{H}, \Lambda) + B\mathcal{D}\left(P\|P_\Lambda\right) + B\sqrt{\frac{1}{2m}\log\frac{|\Lambda|}{\delta}}.$$

The upper bound of the generalization risk on $P$ is mainly determined by its distance to $P_\Lambda$: $\mathcal{D}\left(P\|P_\Lambda\right)$. With the topological prior, risks on $P$ can be tightly bounded by minimizing $\mathcal{D}\left(P\|P_\Lambda\right)$, as long as $P$ falls into the convex hull of training groups. We empirically verify the effectiveness of the topological prior in minimizing the generalization risks over unseen distributions (see Sec. 5).

## 5 EXPERIMENTS

We evaluate TRO in a wide range of tasks including classification, regression, and semantic segmentation. We compare TRO with SOTA baselines on OOD generalization and conduct ablation study on the key components of TRO. Following Gulrajani & Lopez-Paz (2021), we perform model selection based on a validation set constructed from training groups only. We provide implementation details in Appendix 7.2 and results on *DomainBed* (Gulrajani & Lopez-Paz, 2021) in Appendix 7.3.

**Baselines**. We compare TRO with the following methods: (1) Empirical Risk Minimization (ERM) (Vapnik, 1998); (2) Group distributionally robust optimization (DRO) (Sagawa et al., 2019); (3) Invariant Risk Minimization (IRM) (Arjovsky et al., 2019); (4) Risk Extrapolation (REx) (Krueger et al., 2021); (5) Spectral Decoupling (SD) (Pezeshki et al., 2021).

Table 1: Accuracy (%) on *DG-15* and *DG-60*. TRO sets the new SOTA on both *DG-15* and *DG-60*.

|  | ERM | IRM | REx | SD | DRO | TRO (physical) | TRO (data) |
|---|---|---|---|---|---|---|---|
| *DG-15* | 58.00 | 57.87 | 57.22 | 57.56 | 43.22 | 67.56 | **67.89** |
| *DG-60* | 76.02 | 76.61 | 86.89 | 81.04 | 79.59 | 89.19 | **90.72** |

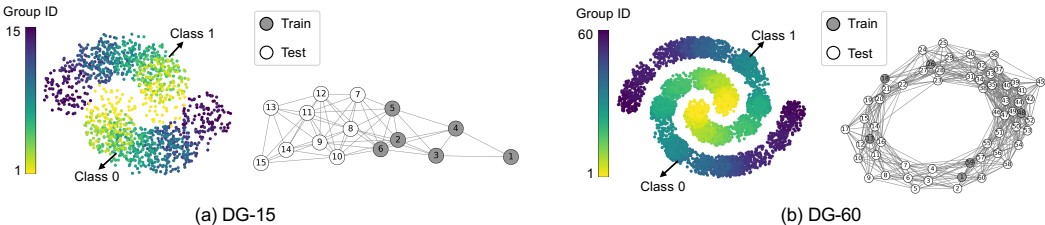

(a) DG-15                           (b) DG-60

Figure 2: Illustration of data groups in (a) *DG-15* and (b) *DG-60* datasets.

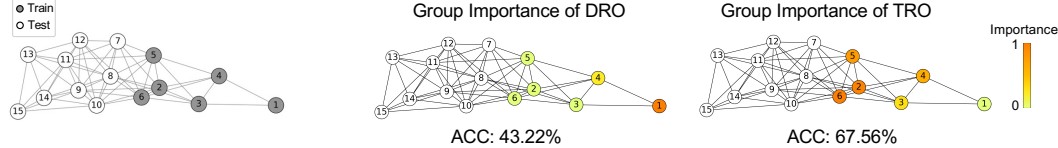

Figure 3: Group importance of DRO and TRO on *DG-15*. DRO assigns the highest weight to the *worst-case* group "*1*" which is the furthest group to the test groups while TRO focuses on the *influential* groups "*2*", "*5*", and "*6*", which are truly connected to test groups.

## 5.1 CLASSIFICATION

**Datasets**. *DG-15* (Xu et al., 2022) is a synthetic binary classification dataset with 15 groups. Each group contains 100 data points. In this dataset, adjacent groups have similar decision boundaries. Following Xu et al. (2022), we use six connected groups as the training groups, and use others as test groups. Note that, different from Xu et al. (2022) which focuses on domain adaptation, the data of test groups are unseen in OOD generalization. *DG-60* (Xu et al., 2022) is another synthetic dataset generated using the same procedure as *DG-15*, except that it contains 60 groups, with 6,000 data points in total. We randomly select six groups as the training groups and use others as test groups. Visualization of *DG-15* and *DG-60* are shown in Fig. 2 (a) and (b), respectively.

**Results**. The results of *DG-15* and *DG-60* are summarized in Tab. 1. In both datasets, our method yields the highest accuracy. For *DG-15*, we show the detailed results of all groups in Fig. 8. We visualize the decision boundary of *DG-15* and *DG-60* in Appendix 7.3.

**Ablations study**. *TRO significantly improves the generalization performance by discovering influential groups*. To investigate the reason why TRO outperforms DRO, we show group weights $\mathbf{q}$ of DRO and TRO on *DG-15* in Fig. 3. DRO assigns the highest weight to group "*1*" which is the furthest group to test groups. Instead, TRO prioritizes influential groups "*2*", "*5*", and "*6*" which are truly connected to the test ones, yielding improved performance on unseen distributions.

## 5.2 REGRESSION

**Datasets**. *TPT-48* (Vose et al., 2014) contains the monthly average temperature for the 48 contiguous states in the US from 2008 to 2019. We focus on the regression task to predict the next 6 months' temperature based on the previous first 6 months' temperature. We consider two generalization tasks: (1) E(24) → W(24): we use the 24 eastern states as training groups and the 24 western states as test groups; (2) N(24) → S(24): we use the 24 northern states as training groups and the 24 southern states as test groups. Test groups one hop away from the closest training group are defined as Hop-1 test groups, those two hops away are Hop-2 test groups, and the remaining groups are Hop-3 test groups. The visualization of N(24) → S(24) on *TPT-48* is shown in Fig. 4 (left).

Table 2: Mean Squared Error (MSE) for both tasks E (24) → W (24) and N (24) → S (24) on *TPT-48*. TRO (data-driven topology) consistently outperforms TRO (physical-based topology) in both tasks, indicating the data-driven topology captures the distributional relation more accurately.

| Task | Group | ERM | IRM | REx | SD | DRO | TRO (physical) | TRO (data) |
|---|---|---|---|---|---|---|---|---|
| E (24)→W (24) | Average of Hop-1 groups | 1.693 | 1.699 | 1.577 | 1.701 | 1.678 | 1.445 | **1.435** |
| | Average of Hop-2 groups | 1.800 | 1.811 | 1.702 | 1.806 | 1.762 | 1.576 | **1.569** |
| | Average of Hop-3 groups | 1.672 | 1.679 | 1.584 | 1.674 | 1.628 | 1.400 | **1.392** |
| | Average of All test groups | 1.716 | 1.724 | 1.616 | 1.722 | 1.684 | 1.466 | **1.458** |
| N (24)→S (24) | Average of Hop-1 groups | 1.084 | 1.133 | **0.487** | 1.169 | 0.931 | 0.889 | 0.855 |
| | Average of Hop-2 groups | 1.265 | 1.312 | **0.944** | 1.354 | 1.170 | 0.991 | 0.950 |
| | Average of Hop-3 groups | 1.975 | 2.021 | 2.266 | 2.091 | 2.027 | 1.678 | **1.604** |
| | Average of All test groups | 1.426 | 1.474 | 1.194 | 1.523 | 1.356 | 1.177 | **1.129** |

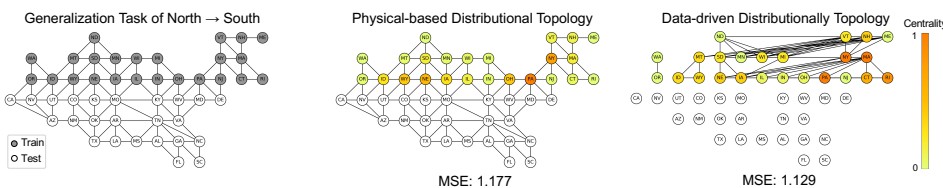

Figure 4: **Left:** Generalization task of North → South on *TPT-48*. **Middle:** Group centrality of physical-based topology. **Right:** Group centrality of data-driven topology. "*PA*" is identified by TRO as the *influential* group in physical-based topology; "*NY*", "*PA*", and "*MA*" are identified by TRO as *influential* groups in data-driven topology. Data topology yields lower MSE than physical topology.

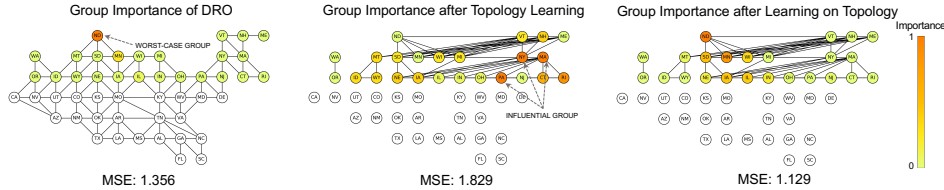

Figure 5: Group importance of DRO and TRO on the North → South task. TRO significantly reduces the generalization risks by not only prioritizing the *worst-case* groups but also the *influential* ones.

**Results**. We show the results of *TPT-48* in Tab. 2. TRO yields the lowest average MSE on both tasks. We also report the average MSE of Hop-1, Hop-2, and Hop-3 test groups for both tasks. Although REx yields the lowest error on Hop-1 and Hop-2 groups in N (24) → S (24), it yields the highest prediction error on Hop-3 groups. The results indicate that REx may yield compromised performance under large distributional drifts. TRO yields the best performance on Hop-3 groups, indicating its strong generalization capability under large distributional drifts.

**Ablations study**. (1) *Data-driven topology yields better performance than physical-based topology.* We show group centrality of both physical and data topology on the task of North → South in Fig. 4. "*PA*" is identified by TRO as the *influential* group in physical-based topology; "*NY*", "*PA*", and "*MA*" are identified by TRO as *influential* groups in data-driven topology. The results prove that the influential groups in data topology are more effective in minimizing the generalization error.

(2) *Strong OOD resilience of TRO comes from the synergy of the worst-case and influential groups.* To investigate which components contribute to the superior performance of TRO. We build a simple baseline based on ERM: we directly use the group importance acquired from the topology to weight training groups and the weights are fixed during the training. We name this baseline as *importance weighted ERM* (IW-ERM). We show the results of "N(24)→S(24)" on *TPT-48* in Tab. 3. The results of IW-ERM are inferior to ERM and DRO, possibly because IW-ERM merely considers influential

Table 3: MSE on *TPT-48*. Ignoring either the *worst-case* (IW-ERM) or *influential* (DRO) groups would yield compromised performance.

| | Hop-1 Avg. | Hop-2 Avg. | Hop-3 Avg. | Avg. |
|---|---|---|---|---|
| ERM | 1.084 | 1.265 | 1.975 | 1.426 |
| IW-ERM | 1.320 | 1.604 | 2.635 | 1.829 |
| DRO | 0.931 | 1.170 | 2.027 | 1.356 |
| TRO | **0.855** | **0.950** | **1.604** | **1.129** |

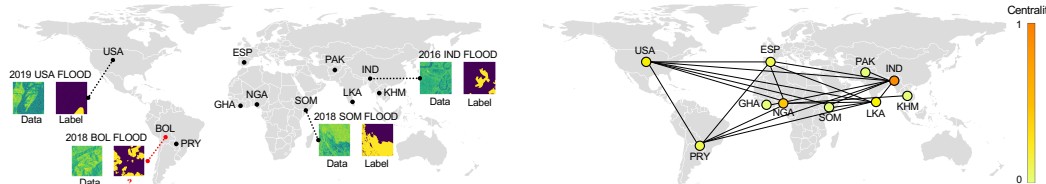

Figure 6: **Left:** Location of the 11 flood events on *Sen1Floods11*. We use the event "*BOL*" for testing and other events for training. **Right:** Data-driven distributional topology on *Sen1Floods11*. (1) "*IND*" and "*NGA*" are identified by TRO as the most influential groups. A possible explanation is that both "*IND*" and "*NGA*" are aroused by heavy rainfall, the most prevalent disaster that causes floods. (2) "*GHA*" and "*KHM*" are identified by TRO as the least influential groups. A possible explanation is that both "*GHA*" and "*KHM*" are aroused edge cases such as dam collapse. The data-driven distributional topology is consistent with domain knowledge and facilitates the explainability of TRO.

groups. We further show the group importance of DRO and TRO in Fig. 5. TRO significantly reduces the generalization risks by not only prioritizing the worst-case groups but also the influential ones.

## 5.3 SEMANTIC SEGMENTATION

**Datasets**. *Sen1Floods11* (Bonafilia et al., 2020) is a public dataset for flood mapping at the global scale. The dataset provides global coverage of 4,831 chips of 512 x 512 10m satellite images across 11 distinct flood events, covering 120,406 $km^2$. Each image is associated with its pixel-wise label. Locations of the 11 flood events are shown in Fig. 6 (left). Flood events vary in boundary conditions, terrain, and other latent factors, posing significant OOD challenges to existing models in terms of reliability and explainability. Following Bonafilia et al. (2020), event "*BOL*" is held out for testing, and data of other events are split into training and validation sets with a random 80-20 split.

|  | ERM | IRM | REx | SD | DRO | TRO (data) |
|---|---|---|---|---|---|---|
| Val | **.489** | .387 | .484 | .449 | .480 | .485 |
| Test | .430 | .338 | .357 | .400 | .433 | **.450** |

Table 4: Segmentation results (IoU) on *Sen1Floods11*. TRO yields better performance than other baselines on unseen flood events.

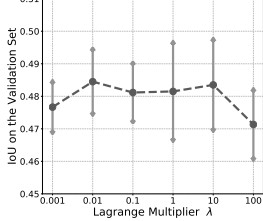

Figure 7: Ablation study on $\lambda$. IoU remains stable for a wide range of $\lambda$.

**Results**. We show the results of *Sen1Floods11* in Tab. 4. ERM achieves the highest IoU on the validation set while TRO achieves the highest IoU on the test set. The results prove that TRO yields better performance than other baselines on unseen flood events.

**Ablations study**. (1) *Data-driven distributional topology is consistent with domain knowledge*. We visualize the distributional topology as well as group centrality in Fig. 6 (right). The learned distributional topology is consistent with domain knowledge, enhancing the explainability of TRO. (2) *Ablation study on $\lambda$*. We report IoU under different $\lambda$ on *Sen1Floods11* in Fig. 7. IoU remains stable for a wide range of $\lambda$, and $\lambda = 0.01$ yields the best performance.

## 6 CONCLUSION

In this paper, we proposed a new principled optimization method that seamlessly integrates topological information to develop strong OOD resilience. Empirical results in a wide range of tasks including classification, regression, and semantic segmentation demonstrate the superior performance of our method over SOTA. Moreover, the data-driven distributional topology is consistent with domain knowledge and facilitates the explainability of our approach.

ACKNOWLEDGEMENTS

This work is partially supported by National Science Foundation (NSF) CMMI-2039857, General University Research (GUR), and University of Delaware Research Foundation (UDRF). The authors would like to thank Kien X. Nguyen for helping with the experiments on DomainBed.

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

# 7 APPENDIX

## 7.1 RELATED WORK

**Distributionally Robust Optimization**. In the context of distributionally robust optimization (DRO), Duchi & Namkoong (2021) and Shalev-Shwartz & Wexler (2016) argued that minimizing the maximal loss over a set of possible distributions can provide better generalization performance than minimizing the average loss. The robustness guarantee of DRO heavily relies on the quality of the uncertainty set which is typically constructed by moment constraints (Delage & Ye, 2010), $f$-divergence (Namkoong & Duchi, 2016) or Wasserstein distance (Shafieezadeh Abadeh et al., 2018). To avoid yielding overly pessimistic models, group DRO (Hu et al., 2018; Sagawa et al., 2019) is proposed to leverage pre-defined data groups to formulate the uncertainty set as the mixture of these groups. Although the uncertainty set of Group DRO is of a wider radius while not being too conservative, our preliminary results show that Group DRO recklessly prioritizes the worst-case groups that incur higher losses than others. Such worst-case groups are not necessarily the influential ones that are truly connected to unseen distributions; optimizing over the worst-case rather than influential groups would yield mediocre OOD generalization performance.

**Out-of-Distribution Generalization**. The goal of OOD generalization is to generalize the model from source distributions to unseen target distributions. There are mainly two branches of methods to tackle OOD generalization: group-invariant learning (Arjovsky et al., 2019; Koyama & Yamaguchi, 2020; Liu et al., 2021) and distributionally robust optimization. The goal of group-invariant learning is to exploit the causally invariant correlations across multiple distributions. Invariant Risk Minimization (IRM) is one of the most representative methods which learns the optimal classifier across source distributions. However, recent work (Rosenfeld et al., 2021) shows that IRM methods can fail catastrophically unless the test data are sufficiently similar to the training distribution.

## 7.2 IMPLEMENTATION DETAILS

In Sec. 3.1, for all hyperparameters such as the kernel scale $\sigma^2$ and the maximum scale $K$, we use the default values from the official implementation[1] of Tong et al. (2021). In Sec. 3.2, for learning rate of model parameters $\eta_\theta$, we use default values from Xu et al. (2022) (*DG-15/-60* and *TPT-48*) and Bonafilia et al. (2020) (*Sen1Floods11*). Therefore, we only tune the learning rate of the mixture distribution $\eta_{\mathbf{q}}$ and the dual variable $\lambda$. All results are reported over 3 random seed runs, which is consistent with Koh et al. (2021) and Shi et al. (2022). We select $\lambda$ from {1e-3, 1e-2, 1e-1, 1, 10, 100} and select $\eta_{\mathbf{q}}$ from {1e-4, 1e-3, 1e-2, 1e-1, 1}.

## 7.3 ADDITIONAL RESULTS

*DG-15* and *DG-60*. We provide detailed classification results for each group. The results are shown in Fig. 8. We can see that, compared to other baselines, TRO significantly improves the generalization performance on groups that are far from the training groups, such as group "*13*". We further visualize the decision boundary of *DG-15* and *DG-60* in Fig. 9 and Fig. 10, respectively.

---

[1]`https://github.com/KrishnaswamyLab/DiffusionEMD`

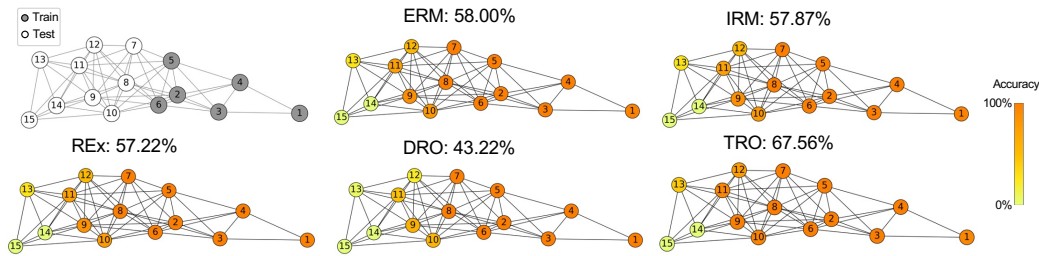

Figure 8: Detailed results on *DG-15*. Our method outperforms ERM by 9.56% while other baselines are inferior to ERM.

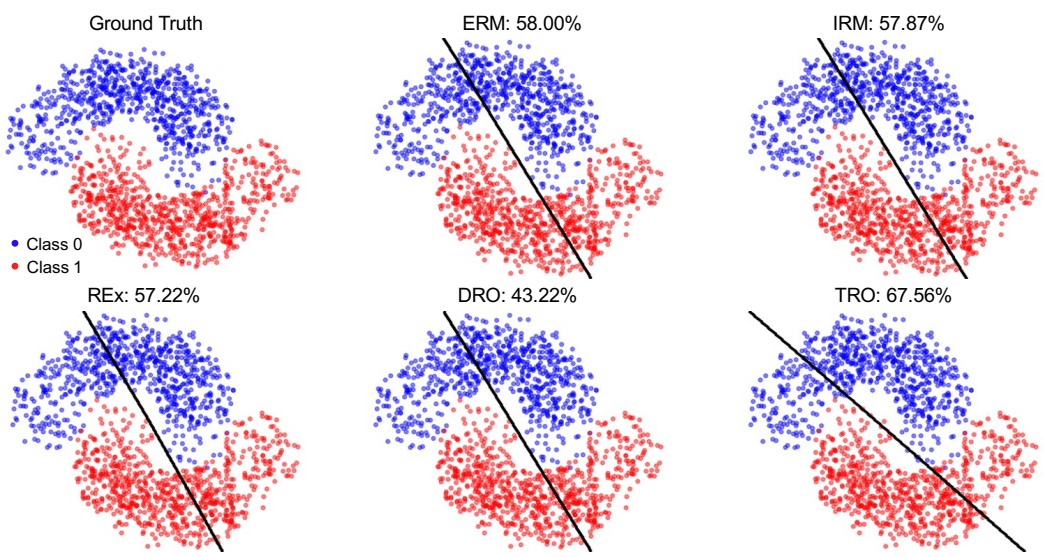

Figure 9: Visualization of decision boundary on *DG-15*.

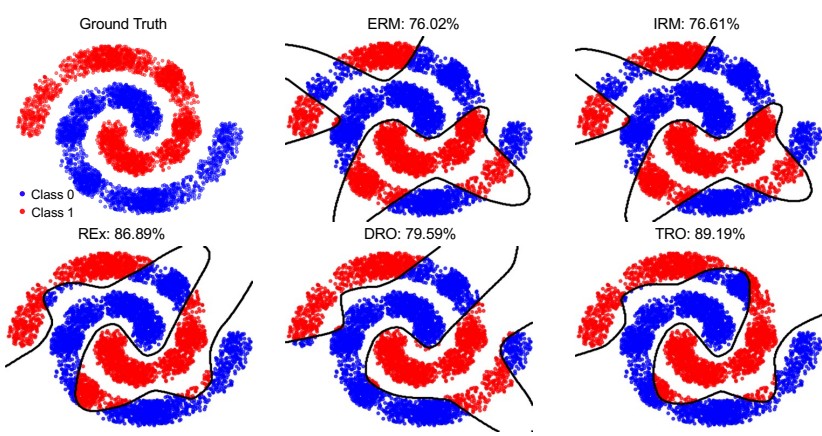

Figure 10: Visualization of decision boundary on *DG-60*. TRO can correctly classify the data of most groups even if training groups are only one-tenth of all groups.

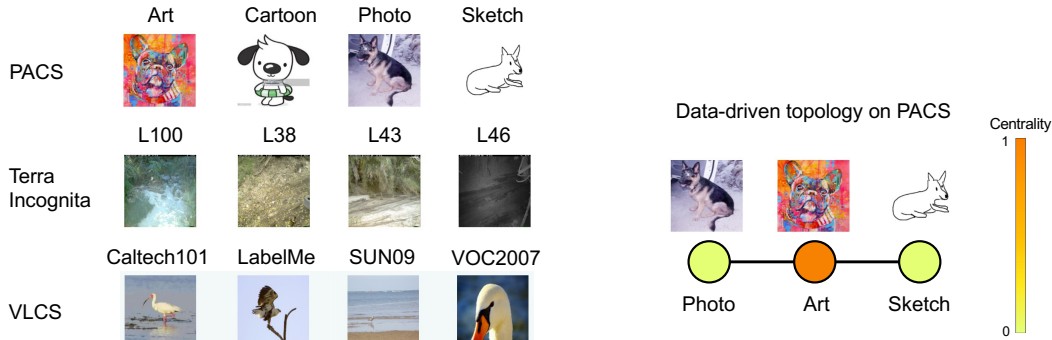

Figure 11: **Left:** Image samples of *DomainBed*. **Right:** the data-driven topology of *PACS* when "*Cartoon*" is the test group while the other three are training groups. We assume the reason why "*Art*" is the most influential group is that "*Art*" may contain more information than "*Photo*" and "*Sketch*" as "*Art*" is the combination of photos and various kinds of styles.

Table 5: Accuracy (%) on *PACS*. "*Art*": "*Art*" is the test group while the other three groups are training groups. In average accuracy, TRO outperforms the SOTA by 0.5% and outperforms ERM and DRO by 0.8% and 2.4%.

|              | Art         | Cartoon     | Photo       | Sketch      | Average     |
|--------------|-------------|-------------|-------------|-------------|-------------|
| ERM          | **88.1**(0.1) | 77.9(1.3)   | 97.8(0)     | 79.1(0.9)   | 85.7(0.5)   |
| Group DRO    | 86.4(0.3)   | 79.9(0.8)   | **98.0**(0.3) | 72.1(0.7)   | 84.1(0.4)   |
| CORAL (SOTA) | 87.7(0.6)   | 79.2(1.1)   | 97.6(0)     | **79.4**(0.7) | 86.0(0.2)   |
| TRO (ours)   | 87.7(0.5)   | **82.1**(0.5) | **98.0**(0.2) | 78.2(1.9)   | **86.5**(0.4) |

Table 6: Accuracy (%) on *Terra*. TRO achieves comparable results with the SOTA and outperforms ERM and DRO by 1.8% and 2.0%.

|              | L100        | L38         | L43         | L46         | Average     |
|--------------|-------------|-------------|-------------|-------------|-------------|
| ERM          | 50.8(1.8)   | 42.5(0.7)   | 57.9(0.6)   | 37.6(1.2)   | 47.2(0.4)   |
| Group DRO    | 47.2(1.6)   | 40.1(1.6)   | 57.6(0.9)   | **43.0**(0.7) | 47.0(0.3)   |
| MMD (SOTA)   | 52.2(5.8)   | **47.0**(0.6) | 57.8(1.3)   | 40.3(0.5)   | **49.3**(1.4) |
| TRO (ours)   | **53.3**(2.4) | 42.2(1.3)   | **59.0**(0.8) | 41.3(0.5)   | 49.0(0.6)   |

*DomainBed*. Following the instructions of the official implementation of *DomainBed* Gulrajani & Lopez-Paz (2021), we have conducted experiments on *PACS* (Li et al., 2017), *Terra* (Beery et al., 2018), and *VLCS* (Fang et al., 2013). Image samples of the three datasets are shown in Fig. 11 (left).

(1) *PACS* is one of the most popular dataset for out-of-distribution generalization. It consists of images from four groups: "*Art*", "*Cartoon*", "*Photo*" and "*Sketch*". Results on *PACS* are shown in Tab. 5. Results of other baselines are from Appendix B.4 of Gulrajani & Lopez-Paz (2021). In average accuracy, TRO outperforms the SOTA by 0.5%. To further investigate the results, we visualize the learned topology in Fig. 11 (right). As observed, when "*Cartoon*" is the test group, the topology is a chain graph consisting of three nodes where "*Art*" is the most influential group. A possible explanation is that "*Art*" may contain more information than "*Photo*" and "*Sketch*" as "*Art*" can be viewed as the combination of photos and various kinds of styles. Even though the topology is so simple, it enables our method to significantly outperforms ERM and DRO by 0.8% and 2.4% on average. The results empirically demonstrate the strong explainability of our method when the number of training groups is quite limited, *i.e.*, 3. We would like to point out that when the distributional shift across different groups is small (see explanation on the results of *VLCS*), the influential group may not exist and all groups share the same centrality. In this special case, TRO aims to strike a good balance between the average (ERM) risk and the worst-case (DRO) risk.

Table 7: Accuracy (%) on *VLCS*. The average accuracy of DRO and TRO is the same. We assume the reason is that the distributional shift across different groups is small (Li et al., 2017), and therefore the influential group may not exist and all groups share the same centrality.

| | Caltech101 | LabelMe | SUN09 | VOC2007 | Average |
|---|---|---|---|---|---|
| ERM | 97.6(1.0) | 63.3(0.9) | 72.2(0.5) | 76.4(1.5) | 77.4(0.3) |
| Group DRO | 97.7(0.4) | 62.5(1.1) | 70.1(0.7) | **78.4**(0.9) | 77.2(0.6) |
| DANN (SOTA) | **98.5**(0.2) | 64.9(1.1) | **73.1**(0.7) | 78.3(0.3) | **78.7**(0.3) |
| TRO (ours) | 96.9(0.2) | **65.0**(0.8) | 71.3(0.9) | 75.5(0.9) | 77.2(0.5) |

(2) *Terra* consists of images of wild animals captured by camera traps under four locations. Results on *Terra* are shown in Tab. 6. Results of other baselines are from Appendix B.6 of Gulrajani & Lopez-Paz (2021). As observed, in average accuracy, TRO achieves comparable results with the SOTA and outperforms ERM and DRO by 1.8% and 2.0%.

(3) Results on *VLCS* are shown in Tab. 7. Results of other baselines are from Appendix B.3 of Gulrajani & Lopez-Paz (2021). The average accuracy of DRO and TRO is the same. We assume the reason is that the distributional shift across different groups is small (Li et al., 2017), and therefore the influential group may not exist and all groups share the same centrality. In this special case, TRO aims to strike a good balance between the average (ERM) risk and the worst-case (DRO) risk. The images of *VLCS* are all photos and the distributional shift is not as significant as *PACS* (*e.g.*, Photo vs. Sketch). As stated in Sec. 2.1 of Li et al. (2017), "despite the famous analysis of dataset bias that motivated the creation of the *VLCS* benchmark, it was later shown that the domain shift is much smaller with recent deep features", and *PACS* (Li et al., 2017) was proposed to address this limitation.

### 7.4 PROOF OF THEOREM 1

*Proof.* By using the property of convex loss functions, we can obtain:

$$\max_{q \in \Delta_m} \mathcal{R}\left(\theta_T, q\right) - \min_{\theta \in \Theta} \mathcal{R}\left(\theta, q_T\right) \leq \frac{1}{T} \max_{q \in \Delta, \theta \in \Theta} \left\{\sum_{t=1}^{T} \mathcal{R}\left(\theta^t, q\right) - \mathcal{R}\left(\theta, q^t\right)\right\},$$

where $\forall t \geq 1$:

$$\mathcal{R}\left(\theta^t, q\right) - \mathcal{R}\left(\theta, q^t\right) = \mathcal{R}\left(\theta^t, q\right) - \mathcal{R}\left(\theta^t, q^t\right) + \mathcal{R}\left(\theta^t, q^t\right) - \mathcal{R}\left(\theta, q^t\right)$$
$$\leq \left\langle\left(q - q^t\right), \nabla_q \mathcal{R}\left(\theta^t, q^t\right)\right\rangle + \left\langle\left(\theta^t - \theta\right), \nabla_\theta \mathcal{R}\left(\theta^t, q^t\right)\right\rangle.$$

By rearranging the terms in the above equation, we obtain:

$$\max_{q \in \Delta, \theta \in \Theta} \left\{\sum_{t=1}^{T} \mathcal{R}\left(\theta^t, q\right) - \mathcal{R}\left(\theta, q^t\right)\right\} \leq \mathbb{E}\left[\max_{\theta \in \Theta} \sum_{t=1}^{T} \left\langle\left(\theta^t - \theta\right), \hat{g}_\theta^t\right\rangle\right] +$$

$$\mathbb{E}\left[\max_{q \in \Delta_m} \sum_{t=1}^{T} \left\langle\left(q - q^t\right), \hat{g}_q^t\right\rangle\right] + \mathbb{E}\left[\max_{q \in \Delta_m} \sum_{t=1}^{T} \left\langle q, \nabla_q \mathcal{R}\left(\theta^t, q^t\right) - \hat{g}_q^t\right\rangle\right]$$

$$+ \mathbb{E}\left[\max_{\theta \in \Theta} \sum_{t=1}^{T} \left\langle\theta, \hat{g}_\theta^t - \nabla_\theta \mathcal{R}\left(\theta^t, q^t\right)\right\rangle\right].$$

Following Collins et al. (2020), we will derive the combined bound by bounding the expectation of each term in the above equation. For the first term, by utilizing the telescoping sum, we can obtain:

$$\mathbb{E}\left[\max_{\theta \in \Theta} \sum_{t=1}^{T} \left\langle\left(\theta^t - \theta\right), \hat{g}_\theta^t\right\rangle\right] = \frac{1}{2} \sum_{t=1}^{T} \frac{1}{\eta_\theta} \left\|\theta^t - \theta\right\|_2^2 + \eta_\theta \left\|\hat{g}_\theta^t\right\|_2^2 - \frac{1}{\eta_\theta} \left\|\theta^t - \eta_\theta \hat{g}_\theta^t - \theta\right\|_2^2$$

$$\leq \frac{2R_\Theta^2}{\eta_\theta} + \frac{\eta_\theta}{2} \sum_{t=1}^{T} \left\|\hat{g}_\theta^t\right\|_2^2 \leq \frac{2R_\Theta^2}{\eta_\theta} + \frac{\eta_\theta T \hat{G}_\theta^2}{2}.$$

Similarly, for the second term:

$$\mathbb{E}\left[\max_{q\in\Delta_m}\sum_{t=1}^{T}\left\langle\left(q-q^t\right),\hat{g}_q^t\right\rangle\right]\leq\frac{2}{\eta_q}+\frac{\eta_q T\hat{G}_q^2}{2}.$$

The third term and the last term are bounded by $\sqrt{T}\tilde{\sigma}_q$ and $R_\Theta\sqrt{T}\tilde{\sigma}_\theta$, respectively. To this end, we can derive the overall bound as:

$$\mathbb{E}\left[\max_{q\in\Delta_m}\mathcal{R}\left(\theta_T,q\right)-\min_{\theta\in\Theta}\mathcal{R}\left(\theta,q_T\right)\right]\leq\frac{2R_\Theta^2}{\eta_\theta T}+\frac{\eta_\theta\hat{G}_\theta^2}{2}+\frac{2}{\eta_q T}+\frac{\eta_q\hat{G}_q^2}{2}+\frac{R_\Theta\tilde{\sigma}_\theta}{\sqrt{T}}+\frac{\tilde{\sigma}_q}{\sqrt{T}}.$$

The above bound can be minimized by setting the step sizes $\eta_\theta=2R_\Theta/\left(\hat{G}_\theta\sqrt{T}\right)$ and $\eta_q=2/\left(\hat{G}_q\sqrt{T}\right)$.

## 7.5 PROOF OF THEOREM 2

*Proof.* Inspired by Qian et al. (2019) and Collins et al. (2020), we utilize the property of M-smooth to start the proof:

$$\mathbb{E}\left[\sum_{i=1}^{n}q_i^t\ell_i\left(\theta^{t+1}\right)\right]\leq\sum_{i=1}^{n}q_i^t\ell_i\left(\theta^t\right)-\left(\eta_\theta-\frac{\eta_\theta^2 M}{2}\right)\left\|g_\theta^t\right\|^2+\frac{\eta_\theta^2 M\sigma_\theta^2}{2}.$$

We rearrange these terms by:

$$\left(\eta_\theta-\frac{\eta_\theta^2 M}{2}\right)\left\|g_\theta^t\right\|^2\leq\mathbb{E}\left[\sum_{i=1}^{n}q_i^t\ell_i\left(\theta^t\right)-\sum_{i=1}^{n}q_i^t\ell_i\left(\theta^{t+1}\right)\right]+\frac{\eta_\theta^2 M\sigma_\theta^2}{2}$$

$$=\mathbb{E}\left[\sum_{i=1}^{n}q_i^t\ell_i\left(\theta^t\right)-\sum_{i=1}^{n}q_i^{t+1}\ell_i\left(\theta^{t+1}\right)\right]$$

$$+\mathbb{E}\left[\sum_{i=1}^{n}q_i^{t+1}\ell_i\left(\theta^{t+1}\right)-\sum_{i=1}^{n}q_i^t\ell_i\left(\theta^{t+1}\right)\right]+\frac{\eta_\theta^2 M\sigma_\theta^2}{2}.$$

The second term of the above equation can be bounded by:

$$\mathbb{E}\left[\sum_{i=1}^{n}q_i^{t+1}\ell_i\left(\theta^{t+1}\right)-\sum_{i=1}^{n}q_i^t\ell_i\left(\theta^{t+1}\right)\right]=\mathbb{E}\left[\sum_{i=1}^{n}\left(q_i^{t+1}-q_i^t\right)\ell_i\left(\theta^{t+1}\right)\right]$$

$$\leq\mathbb{E}\left[\left\|q^{t+1}-q^t\right\|_2\sum_{i=1}^{n}\left(\ell_i\left(\theta^{t+1}\right)\right)^{1/2}\right]$$

$$\leq\eta_q\sqrt{n}\hat{B}\hat{G}_q.$$

By using the Law of Iterated Expectations, we can obtain:

$$\left(\eta_\theta-\frac{\eta_\theta^2 M}{2}\right)\sum_{t=1}^{T}\mathbb{E}\left[\left\|g_\theta^t\right\|^2\right]$$

$$\leq\mathbb{E}\left[\sum_{i=1}^{n}q_i^1\ell_i\left(\theta^1\right)\right]-\mathbb{E}\left[\sum_{i=1}^{n}q_i^{T+1}\ell_i\left(\theta^{T+1}\right)\right]+2T\eta_q\sqrt{n}\hat{B}\hat{G}_q+\frac{TM\eta_\theta^2\sigma_\theta^2}{2}\qquad(9)$$

$$\leq\mathcal{R}\left(\theta^1,q^1\right)+\hat{B}+2T\eta_q\sqrt{n}\hat{B}\hat{G}_q+\frac{T\eta_\theta^2 M\sigma_\theta^2}{2}.$$

Next, we investigate the convergence of $q$:

$$\mathbb{E}\left[\mathcal{R}\left(\theta^t,q\right)-\mathcal{R}\left(\theta^t,q^t\right)\right]=\mathbb{E}\left[\frac{1}{2\eta_q}\left(\left\|q-q^t\right\|_2^2+\left(\eta_q\right)^2\left\|\hat{g}_q^t\right\|_2^2-\left\|q-\left(q^t+\eta_q\hat{g}_q^t\right)\right\|_2^2\right)\right]$$

$$\leq\mathbb{E}\left[\frac{1}{2\eta_q}\left(\left\|q-q^t\right\|_2^2+\left(\eta_q\right)^2\left\|\hat{g}_q^t\right\|_2^2-\left\|q-q^{t+1}\right\|_2^2\right)\right]$$

$$\leq\mathbb{E}\left[\frac{1}{2\eta_q}\left(\left\|q-q^t\right\|_2^2+\left(\eta_q\right)^2\hat{G}_q^2-\left\|q-q^{t+1}\right\|_2^2\right)\right].$$

By aggregating the difference at all time steps, we obtain:

$$\sum_{t=1}^{T} \mathbb{E}\left[\mathcal{R}\left(\theta^t, q\right) - \mathcal{R}\left(\theta^t, q^t\right)\right] \leq \sum_{t=1}^{T} \frac{1}{2\eta_q}\mathbb{E}\left[\left\|q - q^t\right\|_2^2\right] - \frac{1}{2\eta_q}\mathbb{E}\left[\left\|q - q^{t+1}\right\|_2^2\right] + \frac{\eta_q}{2}\hat{G}_q^2$$

$$= \frac{1}{2\eta_q}\mathbb{E}\left[\left\|q - q^1\right\|_2^2\right] + \frac{\eta_q}{2}T\hat{G}_q^2$$

$$\leq \frac{1}{\eta_q} + \frac{\eta_q T\hat{G}_q^2}{2}.$$

Since the above equation holds for all $q \in \Delta_m$, we maximize the right hand side over $q \in \Delta_m$:

$$\frac{1}{T}\sum_{t=1}^{T}\mathbb{E}\left[\mathcal{R}\left(\theta^t, q^t\right)\right] \geq \max_{q \in \Delta_m}\left[\mathcal{R}\left(\theta_T, q\right)\right] - \left(\frac{1}{\eta_q T} + \frac{\eta_q \hat{G}_q^2}{2}\right). \tag{10}$$

Eqs. 9 and 10 show that TRO converges in expectation to an $(\epsilon, \delta)$-stationary point of $\mathcal{R}$ in $\mathcal{O}(1/\epsilon^4)$ stochastic gradient evaluations.

