# OpenReview forum: "Topology-aware Robust Optimization for Out-of-Distribution Generalization"
_ICLR.cc/2023/Conference — ICLR 2023 poster_

### Official Review · Reviewer_bQwH · 2022-10-20

**Confidence:** 4
**Correctness:** 2
**Technical Novelty And Significance:** 3
**Empirical Novelty And Significance:** 3
**Recommendation:** 6

**Clarity, Quality, Novelty And Reproducibility:**

**Clarity.**  The writing is solid, but as noted above, there is insufficient detail to reproduce the algorithm or the experiments.

**Quality.**  I believe that the ideas here are of high quality (as noted above).  However, the quality of the explanation of the main algorithm and of the experiments is lacking.

**Novelty.**  Again, the main idea of incorporating structural priors is novel.  The formulation is not as novel -- it merges the formulation of GroupDRO with the primal-dual scheme used in Model-Based Domain Generalization, as noted by the authors.  The convergence is not novel; the results follow from standard tools.

**Reproducibility.** As noted above, this paper is not reproducible.  Many more details are needed.  This is a major drawback of this paper.

**Strength And Weaknesses:**

### Strengths

**Main idea.**  If this paper is to be accepted, it will be due to the motivation idea, which is that domain generalization is impossible unless some sort of structure is placed on the space of domains.  The argument laid out in the paper -- which is that one must construct a reasonable uncertainty set for the minimax formulation -- is convincing.  Until now, it has been unclear how to define this uncertainty set to engender meaningful notions of OOD generalization.  To this end, imposing structural priors over these domains seems worthwhile, especially when this structure is available a priori.  Momentarily playing devil's advocate, I would suggest that the assumption that the uncertainty set should contain any mixture of training distributions is somewhat strong.  However, on the toy datasets considered at the beginning of the experiments, I would tend to agree that the assumption is more or less realistic.

**Natural optimization-based formulation.**  Grounding this formulation in the minimax formulation of groupDRO also seems natural, as this setting naturally lends itself to restricting the distributions that can be obtained over the training groups.  Furthermore, the analysis from steps (3)-(5) seems natural, as the primal-dual style scheme has been shown to be effective in other domain generalization problems (as noted by the authors).

**Strong empirical results.**  On the datasets that the authors choose, the proposed algorithm yields strong results.  This indicates that structure can help to improve OOD performance.  However, I am concerned that there is insufficient detail regarding reproducibility, which -- if experiments are to be a motivating reason for acceptance -- must be added to the paper (see my comments below).

### Weaknesses

**Failing to satisfy the stated contributions.**  One of the aspects I look for when reviewing is whether or not a paper accomplishes the contributions it lists.  In this regard, I believe that this paper is not successful.  In the last section of the intro, the authors say that one of their contributions is the following:

> "Topology learning methods that are orders of magnitude faster than previous methods to uncover distributional structure from massively collected datasets."

However, the authors do not show -- in the main text or appendix -- any comparison regarding the speed/efficiency of their method.  They allude to the fact that complexity results exist for a particular distance metric (c.f. (Tong et al., 2021) in the paper), but they never (a) explain how these results relate to the method that they propose and (b) they never demonstrate that this holds empirically.  In fact, there is no comparison to any "previous methods" in terms of learning the distribution structure the authors mention, and so the claim quoted above seems to be incorrect as written.

WRT the other contributions, some of them hold, while others are arguable.  The theoretical analysis seems to make a unilateral assumption that the objective R is convex-concave, which as I argue elsewhere in this review, may not hold.  The claims to "explainability" do not seem to be discussed in the experiments, and so it's hard to validate that this claim holds as well.

**Insufficient details for learning the topology.**  Simply put, there are nowhere near enough details given to be able to reproduce the topology learning procedure outlined in Section 3.1.  To begin, the authors do not define what an "affinity matrix" is, or how it is constructed.  As we are presumably computing K on the space of distributions, it's somewhat unclear (at least to me) how it should be computed.  And while some readers will know what an affinity matrix is, others will not.  It's worth explaining in detail here so as to not lose readers.

It's unclear why the notion that "[ERM] captures spurious correlations which preserve group identities" is important in this context.  What do spurious correlations have to do with the problem described thus far?  Does this setting not work for arbitrary distribution shifts?  And moreover, if ERM pretraining is to be performed, how is model selection performed?  Is the pretraining implemented in the same style as the implementation of e.g. IRM in DomainBed?  This should be discussed in the appendix, as the reason for using ERM pretraining is not fundamentally clear.  Moreover, to provide a fair comparison, one would expect the authors to have allowed groupDRO to also use ERM pretraining.  However, this does not seem to be the case.

There are almost no details given on how the Markov diffusion operator P is inferred from K.  At the very least, the authors should devote an appendix to discuss this.  Otherwise, there is no way to reproduce the method.  There simply aren't enough details for anyone to reasonably reproduce this work. And what is $P^t$?  Is t an exponent, or a time index?  The same questions could be asked of $\mu^t$.  And how do these quantities factor into the algorithm? -- Algorithm 1 does not include them.

There should also be further discussion of how one moves from $\mathcal{G}$ -- the graph -- to $p$ -- the structural prior.  There are a few lines on this in the appendix, but as the authors list this as a main contribution, it should be discussed in much more detail in the main text.

Further questions/comments regarding learning the topology:  What does it mean to yield "OOD resilience?"  In what way are these graphical priors consistent with "human knowledge" or "scientific plausibility?"  The authors offer no evidence to back up either of these claims, meaning that these claims of contribution are not verifiable.  And why is the Earth Mover's distance the relevant metric here?

**Convergence analysis.**  The authors seem to assume (unless I've misread their proofs) that $R(f,q)$ is convex-concave.  This seems to not hold, especially as this depends on the distance metric $\mathcal{D}$.  What if $\mathcal{D}$ is not convex?  Then the inner problem seems to have no hope of being concave.  The fact that the analysis relies on this choice of distance metric is not discussed in the paper, and moreover I do not see where the authors mention which distance metric they actually use in the experiments.

Another comment on the convergence analysis is that it does not inform the practical aspects of the algorithm in any way.  The analysis follows from standard tools which are well-known in the literature, and it's not clear what insight this theory brings to the paper.

**Notation.**  I would also argue that the notation in this section -- and more generally, in several parts of the paper -- is unnecessarily confusing.  The authors seem to use the terms "entities," "groups," "domains," and "nodes" to refer to the same thing -- the domains in the domain generalization problem.  The authors do not define $\mathcal{D}^{n_k}$ here.  What is $k$ indexing? The domains?  And what is $\sigma_{k,i}$ vis-a-vis $\sigma$?  Is $\sigma$ a matrix?  And note that $\mathcal{D}$ is overloaded here.  First it is used as a distance metric in (3), and then it is used as a distribution/dataset in Section 4.2.  $d$ is used for the distance in the remainder of the paper, which is confusing.

**Experiments.**  Though the paper reports strong empirical results, there are no details about how the algorithms were tuned.  How many hyperparameter seeds were used?  Was this experiment repeated across multiple trials?  These details are crucial for the paper.

Furthermore, one fundamental drawback is that the authors used somewhat non-standard datasets, in the sense that these datasets are not commonly used in domain generalization papers.  This is fine in general, but the fact that no datasets from e.g. WILDS or DomainBed are used means that as a reader, we have no basis for comparing these results to past work.  The reason that benchmarks like these exist is to provide a metric for measuring progress.  And so by choosing not to use these datasets, it's more difficult for the reader to understand the contribution of this work.

**Summary Of The Paper:**

The main idea in this paper is to incorporate structural priors on the domains/distributions in an out-of-distribution (OOD) generalization setting.  Given a graphical prior over the set of domains, the authors propose a group-DRO-like optimization problem which incorporates this prior by enforcing that the mixture distribution $q$ over groups must be "close" (in some sense) to the prior distribution.  The authors argue that the algorithm they propose for solving this problem enjoys a favorable convergence rate to a saddle point of the minimax formulation.  They also provide experiments concerning classification, regression, and semantic segmentation, showing strong empirical performance vis-a-vis several standard domain generalization baselines.

The meat of the algorithmic contribution involves inferring the structure amongst the domains and imposing that structure on the groupDRO objective.  To this end, the authors consider settings where the structure comprises a graph (G,E), which describes the relationships within the set of domains G by means of the edge weights E.  If the graph structure is known a priori, the authors compute a particular statistic (namely, a distribution over domains/nodes) on the graph, which is used in the robust optimization framework.  Otherwise, if the graph structure is unknown, the authors argue that the relevant statistics can be inferred from data, although a detailed description of *how* this graph is inferred is lacking.  The remaining algorithmic portions of the paper, including the primal-dual-style optimization scheme and the convergence analysis, use standard tools and do not constitute a novel technical innovation.



**Summary Of The Review:**

I thought the main idea here was novel.  I also note that the paper reports strong empirical results, and that the formulation is principled and follows from several well-known results in domain generalization.

On the negative side, the paper is somewhat hard to understand, especially around Section 3.  There are not enough details given to understand or reproduce the topology learning procedure.  The convergence analysis seems to make assumptions that are violated by the formulation, and the analysis does not seem to have any bearing on the practice of domain generalization.  The contributions listed by the authors do not seem to be satisfied.  The experiments also seem to be not reproducible as far as I can tell.

So to summarize, while this paper starts from a novel idea, the execution is lacking.  Reproducibility is a major concern.  This paper could be significantly improved by adding more details about how the relevant structure is learned, and why the authors' method in particular is the right way of doing it.  All of this being said, I feel that in its current form, the paper is not yet ready for publication.  However, I welcome a discussion with the authors.

---

> ### Author Response · Authors · 2022-11-14
> **Response to Reviewer bQwH (1/3)**
>
> We sincerely appreciate your approval of both the motivation and novelty of this work and thank you for constructive suggestions for the improvement of this paper. We have revised our paper according to your suggestions, especially Section 3.1 Topology Learning (all revisions in the manuscript are highlighted in light blue color). We would like to address your remaining concerns as below:
>
> **Q1. More details for learning the topology.**
>
> **Why Empirical Risk Minimization (ERM) is important in topology learning?**
> In topology learning, the main idea is to extract group-specific (rather than group-invariant) features and leverage them to build the distributional topology.
> Take the datasets of DomainBed [1] for example, the rotations of Rotated MNIST dataset and the camera trap locations of Terra Incognita dataset are typical group-specific features.
> Group-specific features are commonly viewed as spurious correlations [2], and recent work [3] shows that ERM recklessly absorbs spurious correlations.
> Inspired by it, we take advantage of the vulnerability of ERM to infer the connectivity between groups.
>
> We train all the models including the ERM-pretrained model using the same selection strategy.
> As stated in Section 5, following [1], we perform model selection based on a validation set constructed from training groups only. In detail, we split each training group into training (80\%) and validation (20\%) subsets. Then, we pool the validation subsets of each training group to create an overall validation set. We will introduce more experimental details in Q3.
>
> Note that the ERM-pretrained model is only used in Section 3.1 topology learning, and we did not use its weights to initialize our model. However, it is undeniable that our method (TRO) uses the information of ERM while Group DRO does not.
> For a more fair comparison, we use the weights of the ERM-pretrained model as the initialization for both TRO and Group DRO.
> The results are shown below (all results in the paper are averaged by three random seed runs as explained in Q3):
>
> |                  | DG-15&uarr; | DG-60&uarr; | TPT-48 (NS)&darr; | TPT-48 (EW)&darr; |
> |------------------|:-----:|:-----:|:-----------:|:-----------:|
> | Group DRO        | 43.22$\pm$0.56 | 79.59$\pm$0.47 |    1.356$\pm$0.053    |    1.684$\pm$0.017    |
> | Group DRO w/ ERM | 57.11$\pm$0.34 | 81.63$\pm$0.79 |    1.321$\pm$0.044    |    1.626$\pm$0.027    |
> | TRO              | **67.89**$\pm$0.79 | 90.72$\pm$0.02 |    1.129$\pm$0.077    |    1.458$\pm$0.024    |
> | TRO w/ ERM       | 67.57$\pm$0.46 | **90.93**$\pm$0.26 |   **1.055**$\pm$0.058    |   **1.402**$\pm$0.019    |
>
> As observed, with the ERM initialization, both Group DRO and TRO yield further improvements on most datasets, while TRO still consistently outperforms Group DRO on all datasets.
>
> **Details on affinity matrix and the Markov diffusion operator.**
> We have added more detailed explanations in the revised paper (see Section 3.1).
> We also revised Algorithm 1 to include more details on topology learning.
>
> **Further discussion of how one moves from $\mathcal{G}$ -- the graph -- to $\mathbf{p}$ -- the structural prior.**
> Thanks for your suggestion. We have moved the discussion from the appendix to the main text and provided more discussion on it.
> Specifically, we have added more discussion on why we use betweenness centrality to measure the centrality of groups:
> Betweenness centrality measures how often an entity is on the shortest path between two other entities in the topology.
> [4] reveals that entities with higher betweenness centrality would have more control over the topology as more information will pass through them.
>
> **Evidence for the claim that these graphical priors are consistent with "human knowledge" or "scientific plausibility".**
> In our experiments on the Sen1Floods11 [5] dataset, we empirically verified this claim by relating the learned topology to the causes of each flood event.
> As stated in the caption of Figure 6, &ldquo;2016 India Flood&rdquo; and &ldquo;2018 Nigeria Flood&rdquo; are identified as the most influential groups, and &ldquo;2015 Gana Flood&rdquo; and &ldquo;2017 Cambodia Flood&rdquo; are the least influential groups. Both &ldquo;2016 India Flood&rdquo; and &ldquo;2018 Nigeria Flood&rdquo; are aroused by heavy rainfall, the most prevalent disaster that causes floods, while &ldquo;2015 Gana Flood&rdquo; and &ldquo;2017 Cambodia Flood&rdquo; are aroused by edge cases such as dam collapse. Therefore, we claim that the topology enjoys strong explainability while being consistent with human knowledge and scientific plausibility.
>
> We will provide more discussion on the Earth Mover's distance in the next question.

---

> > ### Author Response · Authors · 2022-11-14
> > **Response to Reviewer bQwH (2/3)**
> >
> > **Q2. Clarification on the contribution "Topology learning methods that are orders of magnitude faster than previous methods to uncover distributional structure from massively collected datasets.**
> >
> > We apologize for not clearly presenting the contribution. We have revised the statement as "Topology learning methods that are orders of magnitude faster than traditional Earth Mover's distance to uncover distributional structure from massively collected datasets."
> >
> > **Why is the Earth Mover's distance (EMD) the relevant metric in topology learning?**
> > The key idea to learn the topology is to measure the distributional distance between groups (each of which is a collection of data points).
> > Existing literature on learning graphs from data [6] mainly focuses on estimating the distance between data points rather than groups.
> > Earth Mover's distance (EMD) is one of the most popular metrics to measure distances between entire groups as it generalizes the notion of the shortest path between two points to the shortest set of paths between distributions [7].
> > However, the exact computation of EMD is highly expensive.
> > Recently, diffusion EMD [8] was proposed to topologically approximate EMD on the manifold geodesic while cutting down the computational complexity from $O\left(m^2 n^3\right)$ to $\tilde{O}(m n)$ between $m$ distributions over $n$ data points. Therefore, we adopt diffusion EMD as the distance metric in our topology learning.
> >
> > **Difference between our topology learning method and diffusion EMD [8].**
> > The major difference between our topology learning method and the original diffusion EMD [8] is that we leverage the features from an ERM-pretrained model to build the affinity matrix while [8] uses raw data. The merit of ERM has been explained in Q1. Thus our method shares the same computational complexity as [8].
> >
> > **Empirical comparison between EMD and diffusion EMD.**
> > We have conducted preliminary experiments to compare EMD with diffusion EMD in terms of accuracy and efficiency.
> > We ran the experiments on a single NVIDIA GeForce RTX 2080Ti GPU.
> > The results (acc/time(s)) are shown below:
> >
> > |               |    DG-15&uarr;   |    DG-60&uarr;   |  TPT-48 (NS)&darr; |  TPT-48 (EW)&darr; |
> > |---------------|:----------:|:----------:|:------------:|:------------:|
> > | EMD           | **69.23**/6.85 | **91.50**/6.62 | **1.112**/216.70 | **1.449**/224.51 |
> > | Diffusion EMD | 67.89/**0.39** | 90.72/**0.33** |  1.129/**8.27**  |  1.458/**10.28** |
> >
> >
> > As seen, diffusion EMD significantly reduces the computational cost while only suffering from a slight loss in accuracy.
> > The results demonstrate that diffusion EMD strikes a good trade-off between accuracy and efficiency.
> >
> > We would like to clarify that: our paper aims to provide a generic solution on how to utilize the topological information to solve the challenging problem of OOD generalization.
> > Earth Mover's distance is not the exclusive metric for topology learning. Other metrics for distributional distance such as Maximum Mean Discrepancy (MMD) [9] also have the potential to be employed to build the data-driven topology.

---

> > > ### Author Response · Authors · 2022-11-14
> > > **Response to Reviewer bQwH (3/3)**
> > >
> > > **Q3. Details on experiments and results on standard benchmarks such as WILDS or DomainBed.**
> > >
> > > **How the algorithms were tuned?**
> > > In Section 3.1 topology learning, for all hyperparameters such as the kernel scale $\sigma^2$ and the maximum scale $K$, we use the default values from the [official implementation](https://github.com/KrishnaswamyLab/DiffusionEMD) of [8].
> > > In Section 3.2 learning on topology, for learning rate of model parameters $\eta_\theta$, we use default values from [10] (DG-15/-60 and TPT-48) and [5] (Sen1Floods11).
> > > Therefore, we only tune the learning rate of the mixture distribution $\eta_\mathbf{q}$ and the dual variable $\lambda$.
> > > All results are reported over 3 random seed runs, which is consistent with [11] and [12].
> > > In Figure 7, we did show both mean and standard deviation on the validation set of the Sen1Floods11 dataset across different $\lambda$.
> > > We select $\lambda$ from \{1e-3, 1e-2, 1e-1, 1, 10, 100\} and select $\eta_\mathbf{q}$ from \{1e-4, 1e-3, 1e-2, 1e-1, 1\}.
> > > Thanks for pointing it out. We have included the experimental details in the revised paper.
> > >
> > > **Results on standard benchmarks such as WILDS [11] or DomainBed [3].**
> > > In our paper, we conducted experiments on Sen1Floods11 [5] which is a public dataset for global flood mapping during extreme weather events. This dataset is closely related to high-stake and safety-critical applications.
> > > The significance has been recognized by Reviewer uihf: *&ldquo;I believe the algorithm in this paper can potentially have some real-world impact on improving the well-being of humans&rdquo;.*
> > > However, we do agree with the reviewer that "the reason that benchmarks like these exist is to provide a metric for measuring progress".
> > > Due to the tight schedule, it is hard for us to timely deliver the results on the datasets of DomainBed or WILDS with limited computational resources.
> > > We will keep you and other reviewers posted as soon as we get any preliminary results on the two benchmarks.
> > >
> > > **Q4. Clarification on the Distance metric $\mathcal{D}$ in Equation 3 and notations.**
> > >
> > > **Distance metric $\mathcal{D}$.**
> > > $\mathcal{D}$ is usually implemented by $\ell_2$ distance or KL divergence.
> > > First, KL-divergence ($\mathcal{D}_{\mathrm{KL}}(\mathbf{q} \|\| \mathbf{p})=\sum_e q_e \log \left(q_e / p_e\right)$) cannot handle the prior distribution $\mathbf{p}$ with zero elements.
> > > Second, [13] shows that strong convexity is the key to achieve the optimal convergence rate for stochastic gradient optimization.
> > > Therefore, we use $\ell_2$ distance to implement $\mathcal{D}$ due to its strong convexity and simplicity.
> > > When the loss function is convex, $R(f, q)$ is convex-concave. When the loss function is nonconvex, $R(f, q)$ is nonconvex-concave.
> > > We have revised the explanation of $\mathcal{D}$ to make it more clear.
> > >
> > > **Notation.**
> > > Thanks for pointing it out.
> > > In the revised paper, we have carefully addressed the issue of notations by avoiding the introduction of new concepts, providing detailed explanations, and replacing some old notations with new ones.
> > >
> > > We sincerely appreciate your constructive comments and are open to any suggestion that can further improve the quality of our submission.
> > >
> > > [1] Gulrajani et al. "In search of lost domain generalization." ICLR 2021.
> > >
> > > [2] Creager et al. "Environment inference for invariant learning."  ICML 2021.
> > >
> > > [3] Arjovsky et al. "Invariant risk minimization." arXiv 2019.
> > >
> > > [4] Freeman et al. "A set of measures of centrality based on betweenness." Sociometry 1977.
> > >
> > > [5] Bonafilia et al. "Sen1Floods11: A georeferenced dataset to train and test deep learning flood algorithms for sentinel-1." CVPRW 2020.
> > >
> > > [6] Dong et al. "Learning graphs from data: A signal representation perspective." IEEE Signal Processing Magazine 2019.
> > >
> > > [7] Peyré et al. "Computational optimal transport: With applications to data science." Foundations and Trends® in Machine Learning 2019.
> > >
> > > [8] Tong et al. "Diffusion earth mover’s distance and distribution embeddings." ICML 2021.
> > >
> > > [9] Gretton et al. "A kernel two-sample test." JMLR 2012.
> > >
> > > [10] Xu et al. "Graph-Relational Domain Adaptation." ICLR 2022.
> > >
> > > [11] Koh et al. "Wilds: A benchmark of in-the-wild distribution shifts." ICML 2021.
> > >
> > > [12] Shi et al. "Gradient Matching for Domain Generalization." ICLR 2022.
> > >
> > > [13] Rakhlin et al. "Making gradient descent optimal for strongly convex stochastic optimization." ICML 2012.

---

> > > > ### Comment · Reviewer_bQwH · 2022-11-16
> > > > **Rebuttal response**
> > > >
> > > > **Contributions & computational complexity.**  I'm still a bit confused about this.  The revised contribution still doesn't resolve this comment:
> > > >
> > > > > "In fact, there is no comparison to any 'previous methods' in terms of learning the distribution structure the authors mention, and so the claim quoted above seems to be incorrect as written."
> > > >
> > > > And furthermore, I'm still not sure how this is a contribution.  As far as I am aware, the EMD has not been used for constructing graphs for domain generalization.  Therefore, from the reader's perspective, this contribution is effectively saying that: "We are orders of magnitude faster than an algorithm that has never been used or studied for this problem."  Is it the case that given the $O(m^2n^3)$ complexity that it is completely intractable to run this algorithm?  Given than most standard DG benchmarks only have $\leq 6$ domains, this doesn't seem like a huge bottleneck in $m$.
> > > >
> > > > And furthermore, I would push back on the claim that the authors propose these topology learning methods (as stated in the contributions).  While the authors have provided additional details regarding this step, it is still difficult to determine which part of this is a new contribution and which part is due to past work.  It seems more accurate to say that the authors are using previously proposed topology learning methods for domain generalization, which in my opinion constitutes less of a contribution.
> > > >
> > > > **Theoretical results.**  In the rebuttal, the authors say that they use $\mathcal{D}(p,q) = ||p-q||_{L^2}$.  Mathematically, I can see why this is a convenient choice, since it makes the constraint set convex.  However, based on the motivation of this particular paper, I don't understand this choice.  To explain further, consider this sentence:
> > > >
> > > > > "We model the data distributions as many discrete groups lying on a common low-dimensional manifold, where we can explore the distributional topology by either using physical priors or measuring multiscale Earth Mover’s Distance (EMD) among distributions."
> > > >
> > > > However, it seems like in using the $L^2$ norm for $\mathcal{D}$, we lose all of the local information that could potentially be encoded by this prior.  In other words, why should we expect the distributions in an $L^2$ ball around our prior $p$ to remain on the manifold?
> > > >
> > > > Another comment regarding the theory (from the original review):
> > > >
> > > > > "Another comment on the convergence analysis is that it does not inform the practical aspects of the algorithm in any way. The analysis follows from standard tools which are well-known in the literature, and it's not clear what insight this theory brings to the paper."
> > > >
> > > > Could the authors comment on this?
> > > >
> > > > **Missing details.**  I think that adding the additional explanations of how the structural prior is computed (blue text in the updated version) will be helpful for reproducibility.  It is now clear how the algorithm can be run.  However, I'm still a bit concerned about this section.  Although it describes each step in computing $p$, it's still unclear what the intuition is behind each of these steps.  Why is it necessary to normalizing the affinity matrix?  Why is the diffusion operator defined as $P = D^{-1}M$?  And why do we model $X^e$ in this way?
> > > >
> > > > To put it another way, as a reader, this section is still unclear to me because there is a lack of insight into why the algorithm proceeds in this way.  Given the claim that this procedure is a contribution in this paper, it seems reasonable to expect the paper to explain each of the steps in the algorithm, rather than just stating them and then pointing to prior work.
> > > >
> > > > **Notation.**  Thanks for updating the notation.  I think that this will make it easier for readers to understand the main steps.
> > > >
> > > > **Hyperparameters and experiments.**  Regarding the so-called default hyperparameters in past work, it's unclear to me why these defaults should be expected to work more generally.  If someone wanted to reproduce this work, they would then have to dig back through prior work to determine whether the default HPs made sense for their application.
> > > >
> > > > And moreover, since this method is only applied to non-standard datasets, it's somewhat hard to get a sense (empirically) of how the proposed algorithm would stack-up against other algorithms.  I agree that it would be unreasonable to expect the authors to perform these experiments in the time-period of the rebuttal, but I think that this is an essential piece of this paper, without which I would have a hard time justifying acceptance.
> > > >
> > > > This being said, I do think that the ablations that the authors have added will make a stronger case for the paper.  They offer a more complete look at which pieces of the algorithm help (e.g., ERM-pretraining).

---

> > > > > ### Author Response · Authors · 2022-11-23
> > > > > **We would like to hear back from Reviewer bQwH**
> > > > >
> > > > > Dear Reviewer bQwH,
> > > > >
> > > > > In our [latest response](https://openreview.net/forum?id=ylMq8MBnAp&noteId=dwKLAb-Klk3), we have answered your questions and provided results as well as corresponding analysis on DomainBed. We would like to follow up to see if our response addresses your concerns or if you have any further questions.

---

> ### Author Response · Authors · 2022-11-18
> **Response to Reviewer bQwH (1/3, Nov. 18)**
>
> **Q1. Contributions \& computational complexity.**
>
> We would like to clarify that the main contribution of this paper is to leverage the topological structure of distributions to address the problem of out-of-distribution (OOD) generalization.
> We propose a new principled method that seamlessly integrates topological information to distributionally robust optimization (DRO) to develop strong OOD resilience.
> DRO is typically constrained by moment constraints [1], $f$-divergence [2], Wasserstein distance [3] or Maximum Mean Discrepancy [4]. However, how to utilize the topology to constrain DRO still remains a seldom investigated problem.
>
> To eliminate the confusion, we have removed the contribution on topology learning.
>
> >Given that most standard DG benchmarks only have $\leq6$ domains, this doesn't seem like a huge bottleneck in $m$.
>
> We respectfully disagree. In WILDS [5], 8 out of 10 datasets have more than 40 groups (see Figure 2 in [5]).
> The iWildCam dataset has 322 groups, the Amazon data set has 2,586 groups, the Py150 dataset has 8,421 groups, and the OGB-MolPCBA has up to 120,084 groups. These datasets will probably yield a huge computational bottleneck in $m$ (the number of groups).
>
>
> **Q2. Theoretic results.**
>
> >However, it seems like in using the $L^2$ norm for $\mathcal{D}(\mathbf{q} \| \| \mathbf{p})$, we lose all of the local information that could potentially be encoded by this prior. In other words, why should we expect the distributions in an $L^2$ ball around our prior $\mathbf{p}$ to remain on the manifold?
>
> We think the reviewer may have confused the distribution of data with the distribution of group weights.
> We assume the distributions of data lying on a manifold rather than the distributions of group weights.
> In the distance constraint $\mathcal{D}(\mathbf{q} \| \| \mathbf{p})$, $\mathbf{q}$ denotes the distribution over group weight and the prior $\mathbf{p}$ denotes the distribution over group centrality. By minimizing the distance between $\mathbf{q}$ and $\mathbf{p}$, we expect influential groups (groups with high centrality) to be assigned with higher weights than others, such that TRO would strike a good trade-off between the worst-case and influential groups.
> We only need $\mathbf{p}$ to indicate the relative centrality of groups, therefore it is not necessary to encode detailed topological information into it.
> In fact, the design of this prior allows some room for fault tolerance of the topology, which means it is not necessary to learn a precise topology as long as the influential groups (relative centrality of groups) remain unchanged. We will explain more about it in Q4.
>
> >What insight the convergence analysis brings to the paper?
>
> First, the main role of convergence analysis is to prove that the algorithm can converge within finite iterations and therefore guarantee its correctness. If the algorithm does not converge in practice, the reason is most likely that the parameters are not chosen properly while the reason about the algorithm itself can be ruled out.
> Second, the convergence rate serves as a quantitative metric to compare different algorithms in terms of training efficiency.
> Third, the optimal convergence rate can be achieved by setting appropriate learning rates. However, it is nontrivial to estimate the exact learning rate as it requires estimating the bounds of model parameters and gradients. In practice, [6] indicates setting $\eta_\theta \ll \eta_\mathbf{q}$ yields faster convergence for nonconvex-(strongly)-concave optimization.
> In our paper, $\eta_\theta$ is 3e-5 in DG-15/60, and 1e-5 in TPT-48 while $\eta_\mathbf{q}$ is selected from
> \{1e-4, 1e-3, 1e-2, 1e-1, 1\}.
>
> **Q3. More details on topology learning.**
>
> These operations of topology learning follow the standard procedure of diffusion maps [7]. Although we no longer claim this part as our contribution, we still provide more detailed explanation below.
>
> >Why is it necessary to normalize the affinity matrix?
>
> The normalization scales the values such that the range of the row or column values is between 0 and 1, and the normalization makes sure the large values do not overwhelm the smaller values.
>
> >Why is the diffusion operator defined as $\mathbf{P}=\mathbf{D}^{-1} \mathbf{M}$?
>
> This operation is to normalize each row of $\mathbf{M}$ such that the sum of the values in each row is 1, which makes $\mathbf{P}$ a probability matrix.
> $\mathbf{P}_{i, j}$ represents the one-step transition probability from $x_i$ to $x_j$.
>
> >And why do we model $X_e$ in this way?
>
> We assume the reviewer refers to the density estimates  $\mathbf{\mu}_e^t$ for $X_e$.
> Intuitively, $\mathbf{\mu}_e^t$ aggregates the density of group $e$ at time $t$, and it can be viewed as the group embedding at time $t$.
> The group embeddings at different time will be used to measure the distributional distance between groups.

---

> > ### Author Response · Authors · 2022-11-18
> > **Response to Reviewer bQwH (2/3, Nov. 18)**
> >
> > **Q4. Hyperparameters and experiments.**
> >
> > >Why these default hyperparameters should be expected to work more generally.
> >
> > The underlying reason is that our method does not require learning a very precise graph to calculate the prior $\mathbf{p}$ (group centrality) as different graphs may yield the same prior.
> > The hyperparameter $K$ (maximal scale in Equation 3) plays a key role in estimating the diffusion earth mover’s distance.
> > We visualize the graphs and their corresponding group centrality under different $K$.
> > Results are shown in Figure 11 in Appendix 8.3.
> > As observed, although different $K$ yields different graphs, both “NY” and “PA”  are viewed as the most influential groups across all of the graphs.
> > Therefore, there is no significant difference in mean squared error (MSE) among these graphs.
> > In conclusion, our method does not require learning a very precise graph as long as the relative centrality of groups remains stable.
> >
> >
> > [1] Delage et al. "Distributionally robust optimization under moment uncertainty with application to data-driven problems." Operations research 2010.
> >
> > [2] Namkoong, Hongseok, and John C. Duchi. "Stochastic gradient methods for distributionally robust optimization with f-divergences." NeurIPS 2016.
> >
> > [3] Shafieezadeh et al. "Wasserstein distributionally robust Kalman filtering." NeurIPS 2018.
> >
> > [4] Staib, Matthew, and Stefanie Jegelka. "Distributionally robust optimization and generalization in kernel methods." NeurIPS 2019.
> >
> > [5] Koh et al. "Wilds: A benchmark of in-the-wild distribution shifts." ICML 2021.
> >
> > [6] Lin et al. "On gradient descent ascent for nonconvex-concave minimax problems." ICML 2020
> >
> > [7] Coifman, Ronald R., and Stéphane Lafon. "Diffusion maps." Applied and computational harmonic analysis 2006.

---

> > > ### Author Response · Authors · 2022-11-18
> > > **Strong and explainable results on standard benchmark DomainBed (3/3, Nov. 18)**
> > >
> > >
> > >
> > > Following the instructions of the official implementation of DomainBed [1], we have conducted experiments on PACS [2], Terra [3], and VLCS [4]. Image samples of the three datasets are shown in Figure 12 (left) of Appendix 8.3.
> > >
> > > (1) PACS is one of the most popular datasets for out-of-distribution generalization. It consists of images from four groups: "Art", "Cartoon", "Photo" and "Sketch".
> > > Results on PACS are shown below. Results of other baselines are from Appendix B.4 of [1]. "Art": "Art" is the test group while the other three groups are training groups.
> > >
> > > |              |    Art    |  Cartoon  |   Photo   |   Sketch  | Average   |
> > > |--------------|:---------:|:---------:|:---------:|:---------:|-----------|
> > > | ERM          | **88.1**(0.1) | 77.9(1.3) |  97.8(0)  | 79.1(0.9) | 85.7(0.5) |
> > > | Group DRO    | 86.4(0.3) | 79.9(0.8) | **98.0**(0.3) | 72.1(0.7) | 84.1(0.4) |
> > > | CORAL (SOTA) | 87.7(0.6) | 79.2(1.1) | 97.6(0)   | **79.4**(0.7) | 86.0(0.2) |
> > > | TRO (ours)   | 87.7(0.5) | **82.1**(0.5) | **98.0**(0.2) | 78.2(1.9) | **86.5**(0.4) |
> > >
> > > As observed, in average accuracy, TRO outperforms the SOTA by 0.5\%.
> > > To further investigate the results, we visualize the learned topology in Figure 12 (right) of Appendix 8.3.
> > > As observed, when "Cartoon" is the test group, the topology is a chain graph consisting of three nodes where "Art" is the most influential group. We assume the reason is "Art" may contain more information than  "Photo" and "Sketch" as "Art" is the combination of photos and various kinds of styles.
> > > **Even though the topology is so simple, it enables our method to significantly outperforms ERM and DRO by 0.8\% and 2.4\% in average.** The results empirically demonstrate the strong effectiveness and explainability of our method when the number of training groups is quite limited, i.e., 3.
> > >
> > > We would like to point out that when the distributional shift across different groups is small (see explanation on the results of VLCS), the influential group may not exist and all groups share the same centrality. In this special case, TRO aims to strike a good balance between the average (ERM) risk and the worst-case (DRO) risk.
> > >
> > > (2) Terra consists of images of wild animals captured by camera traps under four locations.
> > > Results on Terra are shown below. Results of other baselines are from Appendix B.6 of [1].
> > >
> > > |              |    L100    |  L38  |   L43   |   L46  | Average   |
> > > |--------------|:---------:|:---------:|:---------:|:---------:|-----------|
> > > | ERM          | 50.8(1.8) | 42.5(0.7) |  57.9(0.6)  | 37.6(1.2) | 47.2(0.4) |
> > > | Group DRO    | 47.2(1.6) | 40.1(1.6) | 57.6(0.9) | **43.0**(0.7) | 47.0(0.3) |
> > > | MMD (SOTA) | 52.2(5.8) | **47.0**(0.6) | 57.8(1.3)   | 40.3(0.5) | **49.3**(1.4) |
> > > | TRO (ours)   | **53.2**(2.4) | 42.2(1.3) | **59.0**(0.8) | 41.3(0.5) | 49.0(0.6) |
> > >
> > > As observed, in average accuracy, TRO achieves comparable results with the SOTA, and outperforms ERM and DRO by 1.8\% and 2.0\%.
> > >
> > > (3) VLCS is a combination of four datasets: Caltech101, LabelMe, SUN09, and VOC2007. It consists of photos of five classes: "bird", "car", "chair", "dog", and "person". Results on VLCS are shown below. Results of other baselines are from Appendix B.3 of [1].
> > >
> > > |              |    Caltech101    |  LabelMe  |   SUN09   |   VOC2007  | Average   |
> > > |--------------|:---------:|:---------:|:---------:|:---------:|-----------|
> > > | ERM          | 97.6(1.0) | 63.3(0.9) |  72.2(0.5)  | 76.4(1.5) | 77.4(0.3) |
> > > | Group DRO    | 97.7(0.4) | 62.5(1.1) | 70.1(0.7) | **78.4**(0.9) | 77.2(0.6) |
> > > | DANN (SOTA) | **98.5**(0.2) | 64.9(1.1) | **73.1**(0.7)   | 78.3(0.3) | **78.7**(0.3) |
> > > | TRO (ours)   | 96.9(0.2) | **65.0**(0.8) | 71.3(0.9) | 75.5(0.9) | 77.2(0.5) |
> > >
> > > As observed, the average accuracy of DRO and TRO is the same.
> > > We assume the reason is that the distributional shift across different groups is small [2], and therefore the influential group may not exist and all groups share the same centrality.
> > > In this special case, TRO aims to strike a good balance between the average (ERM) risk and the worst-case (DRO) risk.
> > > The images of VLCS are all photos and the distributional shift is not as significant as PACS (e.g., Photo vs. Sketch).
> > > As stated in Section 2.1 of [2], "Despite the famous analysis of dataset bias that motivated the creation of the VLCS benchmark, it was later shown that the domain shift is much smaller with recent deep features", and PACS [2] was proposed to address this limitation.
> > >
> > > Thank you for your valuable comments. Please let us know if you have any suggestions that can be included in the revised paper.
> > >
> > > [1] Gulrajani, Ishaan, and David Lopez-Paz. "In search of lost domain generalization." ICLR 2021.
> > >
> > > [2] Li et al. "Deeper, broader and artier domain generalization." ICCV 2017
> > >
> > > [3] Beery et al. "Recognition in terra incognita." ECCV 2018
> > >
> > > [4] Fang et al. "Unbiased metric learning: On the utilization of multiple datasets and web images for softening bias." ICCV 2013

---

> > > > ### Comment · Reviewer_bQwH · 2022-11-27
> > > > **More discussion**
> > > >
> > > > Hi authors,
> > > >
> > > > Thanks for you new rebuttal.  I appreciate the time that you've put into addressing the concerns raised in my response.  In the future, if you could respond to my response in the same thread, it would help quite a bit.  It's a bit confusing to dig through multiple threads to try to find the relevant information.
> > > >
> > > > **Contributions.**  I think that removing the claim of novelty on the topology-learning side will clarify the objective of this paper, given that the paper uses existing methods rather than coming up with new methods on that front.
> > > >
> > > > **Datasets.**  WRT this comment:
> > > >
> > > > > "We respectfully disagree. In WILDS [5], 8 out of 10 datasets have more than 40 groups (see Figure 2 in [5]). The iWildCam dataset has 322 groups, the Amazon data set has 2,586 groups, the Py150 dataset has 8,421 groups, and the OGB-MolPCBA has up to 120,084 groups. These datasets will probably yield a huge computational bottleneck in $m$  (the number of groups)."
> > > >
> > > > If you're going to argue that there exist datasets on which the speedup is significant, it's not unreasonable to expect the authors to *demonstrate* this.  The experiments in the paper seem to have 15 and 60 domains respectively.  The domainbed datasets which the authors added have at most 6 domains.  In this regime, I would argue that we shouldn't expect a meaningful speedup.  So while there exist datasets on which one *might* see a speedup, the authors have provided not provided any evidence that backs up this claim empirically.
> > > >
> > > > **Theory.**
> > > >
> > > > WRT this comment:
> > > >
> > > > > "By minimizing the distance between  $p$ and $q$, we expect influential groups (groups with high centrality) to be assigned with higher weights than others, such that TRO would strike a good trade-off between the worst-case and influential groups."
> > > >
> > > > Thanks for the explanation.  It would be worthwhile to clarify this in the text, because this wasn't clear to me when reading the paper originally (although perhaps I simply misunderstood).
> > > >
> > > > With respect to the convergence analysis, I'm still not sure I understand how this analysis will inform the practice of running this algorithm.  For instance, do these assumptions actually hold in practice (especially in the nonconvex case)?  I'm also confused about this sentence:
> > > >
> > > > > "the convergence rate serves as a quantitative metric to compare different algorithms in terms of training efficiency"
> > > >
> > > > Which algorithms are you comparing in terms of training efficiency?  I didn't see the efficiency used as a metric in any part of the paper.  To claim that this is a contribution, it's not enough to simply say that one could use this to compare algorithms.  One must actually show that this algorithm is more efficient than e.g. ERM and GroupDRO in this setting.  This is the essence of what I meant when I originally asked about whether the convergence brought any insight into the procedure of actually training using TRO.
> > > >
> > > > **More details on topology learning.**  These details are important, as without sufficient explanation, the reader will most likely not be able to follow the main argument of the paper.  Adding them will make the overall algorithm more clear in my opinion.
> > > >
> > > > **DomainBed results.**  Adding the comparisons to domainbed is a necessary step, and I appreciate the time that the authors put into readying these results during the rebuttal period.  However, I feel that the empirical results from domainbed actually weaken the paper.  To begin, I would argue that on PACS, TRO does beat SOTA.  SOTA on this dataset (c.f. the table in the README of https://github.com/facebookresearch/DomainBed) is SagNet, which 86.3(0.2).  The result reported here of 86.5(0.4) are marginally larger, although it's unclear if the result is statistically significant.  At best, I would argue that TRO matches SOTA, rather than improving over SOTA by 0.5 as claimed in the rebuttal.  When we compare the results for Terra Incogita to published results, again TRO offers a small but marginal improvement (although as the authors show, MMD actually outperforms TRO when the experiment is rerun).  And on VLCS, DANN outperforms TRO by about one percentage point.
> > > >
> > > > So overall, given that this algorithm doesn't significantly improve over standard baselines on any of the datasets considered, I am not sure that these results constitute an improvement to the paper.  At the end of the day, relative to ERM and DRO, this algorithm is much more complicated to implement and offers perhaps one or two percentage points of improvement over ERM or DRO, which may not be worth it in practice.  On these standard benchmarks, it's therefore not clear that learning the topology in this way leads to significantly superior OOD performance.

---

> > > > > ### Comment · Reviewer_bQwH · 2022-11-27
> > > > > **Continued response**
> > > > >
> > > > > **Final thoughts.**. Throughout this rebuttal process, the authors have put quite a bit of effort into improving this paper.  I believe that many of the changes and clarifications made have strengthened the paper, and therefore it's only fair that I increase my score.  However, I do still have some concerns (as highlighted above) about some aspects of this paper.

---

> > > > > > ### Author Response · Authors · 2022-12-04
> > > > > > **Response to Reviewer bQwH (Dec. 4, 1/2)**
> > > > > >
> > > > > > Dear Reviewer bQwH,
> > > > > >
> > > > > > We would like to thank you for taking the time to go through the rebuttal and for raising the score. We also acknowledge that the discussion with you led to an overall improvement of the paper. We will address your remaining concerns below.
> > > > > >
> > > > > > **Q1. DomainBed results.**
> > > > > >
> > > > > > We thank the reviewer for bringing to our attention the table in the [README](https://github.com/facebookresearch/DomainBed) of DomainBed [1]. The results of other baselines in Tables. 5-7 of our paper are from the reported results of the [paper](https://arxiv.org/pdf/2007.01434.pdf). According to the README, on Terra [2], the result of MMD (SOTA in the paper) is 42.2\% and the result of SagNet (SOTA in the README) is 48.6\%, which is lower than 49.0\% (our method TRO). On VLCS [3], although CORAL yields the highest accuracy 78.8\%, the results of CORAL on PACS [4] and Terra are inferior to ours: 86.2\% vs. 86.5\% on PACS and 47.6\% vs. 49.0\% on Terra. In addition, even the recent representative methods on OOD generalization such as [Fish](https://openreview.net/pdf?id=vDwBW49HmO) [5] cannot significantly outperform ERM on DomainBed. We summarize the results below.
> > > > > >
> > > > > > |            | PACS | Terra | VLCS | Average |
> > > > > > |------------|:----:|:-----:|:----:|---------|
> > > > > > | ERM        | 85.5 |  46.1 | 77.5 |   69.7  |
> > > > > > | Fish       | 85.5 | 45.1  | **77.8** |   69.5  |
> > > > > > | TRO (ours) | **86.5** | **49.0**  | 77.2 |   **70.9**  |
> > > > > >
> > > > > > **First**, we would like to point out that accuracy is not the only metric to evaluate OOD generalization: as many scientific applications involve high-regret decision-making processes, it is equally essential to provide reliable model explanations. The lack of explainability would impede the safe deployment of models on unseen distributions. We empirically show the explainability of TRO on Sen1Floods11 [6] (see Section 5.3 & Figure 6) and DomainBed (see Appendix 8.3 & Figure 12).
> > > > > >
> > > > > > **Second**, although DomainBed streamlines rigorous and reproducible experimentation in OOD generalization, it is still not the "perfect" benchmark (as we stated in the [general response](https://openreview.net/forum?id=ylMq8MBnAp&noteId=LpF4HT5J1uY)):
> > > > > > 1) DomainBed only has a single task: image classification, while our experiments cover classification, regression, and semantic segmentation.
> > > > > > We use the same model selection strategy as DomainBed did, and all baselines and their implementations are also from DomainBed.
> > > > > > 2) The datasets of DomainBed are not related to safety-critical scenarios, while Sen1Floods11, a public dataset adopted in our experiments, is closely related to high-stake applications.
> > > > > >
> > > > > > **Third**, our goal is not to propose another benchmark for OOD generalization.
> > > > > > Instead, we aim to pave the way for leveraging the topological structure of distributions to perform OOD generalization according to the graph, rather than blindly generalize to unseen distributions.
> > > > > >
> > > > > > **Q2. More discussion on topology learning (Section 3.1).**
> > > > > >
> > > > > > >Relative to ERM and DRO, this algorithm is much more complicated to implement and offers perhaps one or two percentage points of improvement over ERM or DRO, which may not be worth it in practice.
> > > > > >
> > > > > > As stated in our previous [response](https://openreview.net/forum?id=ylMq8MBnAp&noteId=J9A6fH6fX5), our paper aims to provide a generic solution on how to utilize the topological information to solve the challenging problem of OOD generalization. Earth Mover's distance (EMD) is not the exclusive metric for topology learning.
> > > > > > In our preliminary experiments, we have tried a much simpler method for topology learning: if we assume the features of each data group follow multivariate Gaussian distributions: $f(X_e)\sim\mathcal{N}(\mu_e, \Sigma_e)$, the 2-Wasserstein ($W_2$) distance between $f(X_e)$ and $f(X_{e^\prime})$ has a [closed-form solution](https://djalil.chafai.net/blog/2010/04/30/wasserstein-distance-between-two-gaussians/).
> > > > > > We summarize the results (acc/time(s)) of all distributional distances mentioned in the rebuttal.
> > > > > >
> > > > > > |               |    DG-15&uarr;   |    DG-60&uarr;   |  TPT-48 (NS)&darr; |  TPT-48 (EW)&darr; |
> > > > > > |---------------|:----------:|:----------:|:------------:|:------------:|
> > > > > > | EMD           | **69.23**/6.85 | **91.50**/6.62 | **1.112**/216.70 | **1.449**/224.51 |
> > > > > > | $W_2$(Guassian) | 63.37/**7e-4** | 87.92/**7e-4** | 1.203/**2e-3** | 1.491/**2e-3** |
> > > > > > | Diffusion EMD | 67.89/0.39 | 90.72/0.33 |  1.129/8.27 |  1.458/10.28 |
> > > > > >
> > > > > > As observed, although $W_2$(Gaussian) yields lower accuracy than EMD and diffusion EMD, it still outperforms ERM and DRO by 5.37%/11.9% and 20.15%/8.33% on DG-15/60.
> > > > > >
> > > > > > **To summarize**:
> > > > > > 1) If accuracy is the priority while the computational cost is not the major concern (such as DomainBed which only has $\leq$ 6 groups, as pointed out by the reviewer), EMD would be preferred.
> > > > > > 2) If efficiency and simplicity are the priority, $W_2$(Gaussian) would be preferred.
> > > > > > 3) If a good trade-off between accuracy and efficiency is the priority, diffusion EMD would be preferred.

---

> > > > > > > ### Author Response · Authors · 2022-12-04
> > > > > > > **Response to Reviewer bQwH (Dec. 4, 2/2)**
> > > > > > >
> > > > > > > **Q3. Theory**
> > > > > > >
> > > > > > > > Which algorithms are you comparing in terms of training efficiency?
> > > > > > >
> > > > > > > As we cast OOD generalization as a worst-case problem, in terms of convergence analysis, it would be more fair to compare with other algorithms based on minimax optimization.
> > > > > > > The assumptions such as Lipschitz continuity and smoothness are commonly adopted in existing literature [7][8].
> > > > > > >
> > > > > > > **Q4. Datasets**
> > > > > > >
> > > > > > > > The experiments in the paper seem to have 15 and 60 domains respectively.
> > > > > > >
> > > > > > > The total numbers of domains (groups) in DG-15/60 are 15/60, but we only use 6 domains for training and use the remaining domains for testing. Indeed, compared to DomainBed, WILDS [9] would be a more suitable benchmark to evaluate the efficiency of TRO. We will leave it as future work.
> > > > > > >
> > > > > > > Thank you again for continually helping us improve the quality of the paper. We will incorporate all the suggestions to the final version.
> > > > > > >
> > > > > > > [1] Gulrajani et al. "In search of lost domain generalization." ICLR 2021.
> > > > > > >
> > > > > > > [2] Beery et al. "Recognition in terra incognita." ECCV 2018.
> > > > > > >
> > > > > > > [3] Fang et al. "Unbiased metric learning: On the utilization of multiple datasets and web images for softening bias." ICCV 2013.
> > > > > > >
> > > > > > > [4] Li et al. "Deeper, broader and artier domain generalization." ICCV 2017.
> > > > > > >
> > > > > > > [5] Shi et al. "Gradient Matching for Domain Generalization." ICLR 2022.
> > > > > > >
> > > > > > > [6] Bonafilia et al. "Sen1Floods11: A georeferenced dataset to train and test deep learning flood algorithms for sentinel-1." CVPRW 2020.
> > > > > > >
> > > > > > > [7] Lin et al. "On gradient descent ascent for nonconvex-concave minimax problems." ICML 2020.
> > > > > > >
> > > > > > > [8] Jin et al. "Non-convex distributionally robust optimization: Non-asymptotic analysis." NeuIPS 2021.
> > > > > > >
> > > > > > > [9] Koh et al. "Wilds: A benchmark of in-the-wild distribution shifts." ICML 2021.

---

> > > > > > > > ### Comment · Reviewer_bQwH · 2022-12-08
> > > > > > > > **More discussion**
> > > > > > > >
> > > > > > > > Regarding the **First** section, I tend to agree with the authors here.  The explainability of this method is a notable benefit over other methods, and I think that perhaps I overlooked this in my initial assessment.  The **Second** and **Third** points are also valid in my opinion.  Looking back over our responses, I think the message of the paper has become clearer -- to make progress in domain generalization, we must utilize the structure that exists between the domains.  And this paper is the first that I've seen that attempts to do this in a principled way.  I will bump my score again, because I think that the novelty of this paper is strong, and that this novelty has become clearer over the course of the rebuttal.

---

> > > > > > > > > ### Author Response · Authors · 2022-12-09
> > > > > > > > > **Thank you for your support**
> > > > > > > > >
> > > > > > > > > Dear Reviewer bQwH,
> > > > > > > > >
> > > > > > > > > Thank you for acknowledging the novelty of our paper and voting for acceptance. We also appreciate your comments regarding the review of Reviewer BJCq. We are fortunate to have reviewers like you in this community.
> > > > > > > > >
> > > > > > > > > Best,
> > > > > > > > >
> > > > > > > > > Paper408 Authors

---

### Official Review · Reviewer_zKvZ · 2022-10-20

**Confidence:** 1
**Correctness:** 2
**Technical Novelty And Significance:** 2
**Empirical Novelty And Significance:** Not applicable
**Recommendation:** 5

**Clarity, Quality, Novelty And Reproducibility:**

--

**Strength And Weaknesses:**

--

**Summary Of The Paper:**

--

**Summary Of The Review:**

--

---

### Official Review · Reviewer_vFEm · 2022-10-21

**Confidence:** 3
**Correctness:** 3
**Technical Novelty And Significance:** 3
**Empirical Novelty And Significance:** 3
**Recommendation:** 6

**Clarity, Quality, Novelty And Reproducibility:**

This paper is moderate in clarity and quality. The proposed topology-based DRO is novel enough. Implementation details and codes are provided to ensure reproducibility.

**Details Of Ethics Concerns:**

No ethic concerns appear.

**Strength And Weaknesses:**

Strength:
* As traditional DRO methods are only based on the loss values, utilization of the topological knowledge can further improve the uncertainty set selection, thus can improve the learning performance.
* The experiments are pretty sufficient, both quantitative and qualitative evidence is provided to support the proposed TRO. Moreover, three tasks including classification, regression, and semantic segmentation are investigated to show the effectiveness of TRO.
* The proposed TRO is theoretically guaranteed.

Weakness:
* The writing of this paper can be further polished. There are many unclear sentences and unprofessional writing styles. For example, what is the “$L^1$ distance between two groups”? Since $\ell_2$ is used in this paper, is it supposed to be “$\ell_1$ distance”? Moreover, it is suggested to explain the notations $q_e$ and $P_e$ to avoid misunderstanding.
* The three steps of data-driven topology are not clearly presented and it is quite difficult to understand for unfamiliar readers. These three steps seem to be isolated from each other, I cannot see any relevance between them and how they are conducted to produce the topology. Could you please give a more detailed explanation?
* The constraint in Eq. (3) seems to be erroneously motivated. The distance constraint between $\mathbf{p}$ and $\mathbf{q}$ can only make sure that they are numerically close to each other, instead of having a similar topology structure. Intuitively, as $\mathbf{p}$ tries to find the most influential groups, and $\mathbf{q}$ focus on finding the groups with large losses, so the optimal $\mathbf{q}$ is the groups that have both large loss and influence. Is this a correct interpretation?
* More importantly, the proposed method is a two-stage learning process. The computational cost of learning the topology is not provided.


**Summary Of The Paper:**

This paper studies the distributionally robust optimization problem by leveraging topological knowledge. Specifically, two types of topologies are considered, namely physical-based topology and data-driven topology. The physical-based topology is based on neighborhood information, and the data-driven topology is based on the affinity matrix of the data. By constructing the uncertainty set based on one of the considered topologies, the proposed TRO method is shown to be superior to many popular methods.

**Summary Of The Review:**

I have carefully read the methodology and experiments. This paper managed to make some contributions to DRO, however, there are still some concerns (see weaknesses). If the authors can address my concerns, I will consider raising my score.

---

> ### Author Response · Authors · 2022-11-14
> **Response to Reviewer vFEm**
>
> Thanks for your valuable comments. We are glad that you find our idea is ```"novel enough"```, our experiments ```"are pretty sufficient"```, and ```"implementation details and codes are provided to ensure reproducibility"```. Below we address your questions one by one.
>
> **Q1. Clarification on notations.**
>
> In the revised paper, we have replaced the notion of $L^1$ distance with $\ell_1$ distance and added the explanation for $q_e$ and $P_e$: $q_e$ denotes the weight of group $e$ and $P_e$ denotes the distribution of group $e$.
>
> Additionally, we perform an exhaustive re-examination of the paper to ensure that the notions are consistent and clear.
>
> **Q2. More details on the three steps of data-driven topology.**
>
> We have revised the three steps of data-driven topology with more details (see Section 3.1). We also revised Algorithm 1 to include more details on topology learning.
>
> **Q3. The distance constraint between $\mathbf{p}$ and $\mathbf{q}$ can only make sure that they are numerically close to each other, instead of having a similar topology structure. Intuitively, as $\mathbf{p}$ tries to find the most influential groups, and $\mathbf{q}$ focus on finding the groups with large losses, so the optimal $\mathbf{q}$ is the groups that have both large loss and influence. Is this a correct interpretation?**
>
> Thanks for pointing it out. Your interpretation is correct. We have revised the intuitive explanation for Equation 3.
>
> **Q4. The computational cost of learning the topology.**
>
> The computational complexity of learning the topology is $\tilde{O}(mn)$ between $m$ distributions over $n$ data points, as stated in Section 3.1.
>
> Again, thank you for your valuable comments. Please do let us know if you have any remaining questions or comments.

---

> > ### Comment · Reviewer_vFEm · 2022-11-16
> > **Experimental Results**
> >
> > Thanks for addressing all my concerns.
> >
> > However, I am wondering if the experimental results are correct since the original **distance constraint** was wrong. In my opinion, the distance constraint is an essential part of the topology, which might lead to significant performance differences if it is changed. Have you re-run the experiments?

---

> > > ### Author Response · Authors · 2022-11-16
> > > **Response to Reviewer vFEm**
> > >
> > > Dear Reviewer vFEm,
> > >
> > > Thanks for your prompt reply.
> > >
> > > We would like to clarify that the motivation of the **distance constraint** in Equation 4 (Equation 3 in the original paper) is right.
> > > What is wrong is the intuitive explanation of Equation 4, which is the sentence immediately following the equation. We highlighted the sentence in light blue color in the revised paper. Therefore, the revision of this sentence does not affect either the formulation of the distance constraint nor the experimental results.
> > >
> > > Please let us know if our response addresses your concerns or if you have any further questions.

---

> > > > ### Comment · Reviewer_vFEm · 2022-11-16
> > > > **Concerns Addressed**
> > > >
> > > > Thanks for your clarification. All my concerns are addressed, I have raised my rating to 6, and voted for acceptance.

---

> > > > > ### Author Response · Authors · 2022-11-16
> > > > > **Thanks for your support**
> > > > >
> > > > > Thank you for acknowledging the novelty of our work and raising your score! We appreciate your efforts in all the constructive suggestions and discussions that help to improve this paper.

---

> ### Author Response · Authors · 2022-11-18
> **Strong and explainable results on standard benchmark DomainBed**
>
> Dear Reviewer vFEm,
>
> As suggested by Reviewer bQwH, we conducted experiments on the standard benchmark DomainBed [1]. Following the instructions of the official implementation of DomainBed [1], we have conducted experiments on PACS [2], Terra [3], and VLCS [4]. Image samples of the three datasets are shown in Figure 12 (left) of Appendix 8.3.
>
> (1) PACS is one of the most popular datasets for out-of-distribution generalization. It consists of images from four groups: "Art", "Cartoon", "Photo" and "Sketch".
> Results on PACS are shown below. "Art": "Art" is the test group while the other three groups are training groups.
>
> |              |    Art    |  Cartoon  |   Photo   |   Sketch  | Average   |
> |--------------|:---------:|:---------:|:---------:|:---------:|-----------|
> | ERM          | **88.1**(0.1) | 77.9(1.3) |  97.8(0)  | 79.1(0.9) | 85.7(0.5) |
> | Group DRO    | 86.4(0.3) | 79.9(0.8) | **98.0**(0.3) | 72.1(0.7) | 84.1(0.4) |
> | CORAL (SOTA) | 87.7(0.6) | 79.2(1.1) | 97.6(0)   | **79.4**(0.7) | 86.0(0.2) |
> | TRO (ours)   | 87.7(0.5) | **82.1**(0.5) | **98.0**(0.2) | 78.2(1.9) | **86.5**(0.4) |
>
> As observed, in average accuracy, TRO outperforms the SOTA by 0.5\%.
> To further investigate the results, we visualize the learned topology in Figure 12 (right) of Appendix 8.3.
> As observed, when "Cartoon" is the test group, the topology is a chain graph consisting of three nodes where "Art" is the most influential group. We assume the reason is "Art" may contain more information than  "Photo" and "Sketch" as "Art" is the combination of photos and various kinds of styles.
> **Even though the topology is so simple, it enables our method to significantly outperforms ERM and DRO by 0.8\% and 2.4\% in average.** The results empirically demonstrate the strong effectiveness and explainability of our method when the number of training groups is quite limited, i.e., 3.
>
> We would like to point out that when the distributional shift across different groups is small (see explanation on the results of VLCS), the influential group may not exist and all groups share the same centrality. In this special case, TRO aims to strike a good balance between the average (ERM) risk and the worst-case (DRO) risk.
>
> (2) Terra consists of images of wild animals captured by camera traps under four locations.
> Results on Terra are shown below.
>
> |              |    L100    |  L38  |   L43   |   L46  | Average   |
> |--------------|:---------:|:---------:|:---------:|:---------:|-----------|
> | ERM          | 50.8(1.8) | 42.5(0.7) |  57.9(0.6)  | 37.6(1.2) | 47.2(0.4) |
> | Group DRO    | 47.2(1.6) | 40.1(1.6) | 57.6(0.9) | **43.0**(0.7) | 47.0(0.3) |
> | MMD (SOTA) | 52.2(5.8) | **47.0**(0.6) | 57.8(1.3)   | 40.3(0.5) | **49.3**(1.4) |
> | TRO (ours)   | **53.2**(2.4) | 42.2(1.3) | **59.0**(0.8) | 41.3(0.5) | 49.0(0.6) |
>
> As observed, in average accuracy, TRO achieves comparable results with the SOTA, and outperforms ERM and DRO by 1.8\% and 2.0\%.
>
> (3) VLCS is a combination of four datasets: Caltech101, LabelMe, SUN09, and VOC2007. It consists of photos of five classes: "bird", "car", "chair", "dog", and "person". Results on VLCS are shown below.
>
> |              |    Caltech101    |  LabelMe  |   SUN09   |   VOC2007  | Average   |
> |--------------|:---------:|:---------:|:---------:|:---------:|-----------|
> | ERM          | 97.6(1.0) | 63.3(0.9) |  72.2(0.5)  | 76.4(1.5) | 77.4(0.3) |
> | Group DRO    | 97.7(0.4) | 62.5(1.1) | 70.1(0.7) | **78.4**(0.9) | 77.2(0.6) |
> | DANN (SOTA) | **98.5**(0.2) | 64.9(1.1) | **73.1**(0.7)   | 78.3(0.3) | **78.7**(0.3) |
> | TRO (ours)   | 96.9(0.2) | **65.0**(0.8) | 71.3(0.9) | 75.5(0.9) | 77.2(0.5) |
>
> As observed, the average accuracy of DRO and TRO is the same.
> We assume the reason is that the distributional shift across different groups is small [2], and therefore the influential group may not exist and all groups share the same centrality.
> In this case, TRO aims to strike a good balance between the average (ERM) risk and the worst-case (DRO) risk.
> The images of VLCS are all photos and the distributional shift is not as significant as PACS (e.g., Photo vs. Sketch).
> As stated in Section 2.1 of [2], "Despite the famous analysis of dataset bias that motivated the creation of the VLCS benchmark, it was later shown that the domain shift is much smaller with recent deep features", and PACS [2] was proposed to address this limitation.
>
> Thank you for your valuable comments. Please let us know if you have any suggestions that can be included in the revised paper.
>
> [1] Gulrajani, Ishaan, and David Lopez-Paz. "In search of lost domain generalization." ICLR 2021.
>
> [2] Li et al. "Deeper, broader and artier domain generalization." ICCV 2017
>
> [3] Beery et al. "Recognition in terra incognita." ECCV 2018
>
> [4] Fang et al. "Unbiased metric learning: On the utilization of multiple datasets and web images for softening bias." ICCV 2013

---

### Official Review · Reviewer_BJCq · 2022-10-24

**Confidence:** 5
**Correctness:** 2
**Technical Novelty And Significance:** 1
**Empirical Novelty And Significance:** 1
**Recommendation:** 3

**Clarity, Quality, Novelty And Reproducibility:**

* **Clarity**
  - This paper is well-written and easy to follow.

* **Quality**
  - The contributions are all over-claimed.

* **Novelty**
  - The novelty is limited.

* **Reproducibility**
  -  The study reported in sufficient detail to allow for its reproducibility.



**Details Of Ethics Concerns:**

None.

**Strength And Weaknesses:**

* **Strength**

  - Learning the group centrality to construct the uncertainty set explores the relationship of the training data.


* **Weakness**

  - The novelty is limited to combining the existing methods (graph construction, group centrality, optimization method & convergence rate, generalization bound).

  - The definition of uncertainty set is incorrect.

**Summary Of The Paper:**

Compared to the standard distributionally robust optimization (DRO, Namkoong & Duchi 2016), this paper constructs a new uncertainty set. The uncertainty set of DRO is an f-divergence ball whose center is the uniform distribution of all training samples. The new uncertainty set is a distributional metric ball whose center is the group centrality of the graph constructed by training data. This graph-based uncertainty set captures the correlation of samples. Then solving the corresponding problem can utilize the off-the-shelf method. The theoretical analyses are similar to the existing results in the DRO and ML communities. When the graph captures the underlying relation of training data, the proposed method shows promising results.

**Summary Of The Review:**

This paper introduces the group centrality to construct the uncertainty set. This method is applicable to distributionally robust optimization for graph data. However, the novelty is limited to combining the existing methods (graph construction, group centrality, optimization method & convergence rate, generalization bound). Consequently, it does not reach the requirement of ICLR.

---

> ### Comment · Reviewer_bQwH · 2022-11-07
> **Regarding your review**
>
> Hello -- I'm also a reviewer on this paper.  Regarding your review, I see that you think this paper should be rejected.  However, it's unclear to me -- and I'm sure it's also unclear to the authors -- why you came to this conclusion.  Below, I explain my confusion, and based on this I hope that the reviewer will consider providing more detail in their review.
>
> Based on your review, it seems that you have two concerns.
>
> First, you believe that the paper is limited to combining existing methods.  However, you do not offer any explanation regarding why you feel that this is a weakness.  Indeed, I would argue that the majority of papers written in ML conferences can be viewed as combining existing methods.  For example, the paper *Assessing Generalization of SGD via Disagreement* (https://openreview.net/forum?id=WvOGCEAQhxl), which received a spotlight at last year's ICLR after receiving four 8s, does not propose any new theoretical tools.  Instead, it details the discovery of a new phenomenon, and then uses various existing frameworks (e.g., optimization, statistics, etc.) in an attempt to explain this phenomenon.  And despite this, all four reviewers agreed that the paper had high novelty.  So while in principle I agree that the paper under review combines existing methods, this argument on its own does not seem sufficient to conclude that the paper lacks novelty.
>
> Second, you believe that there is an error regarding the definition of the uncertainty set.  However, you do not explain what this error is, and you do not comment on how this potential impacts the rest of the paper.  This puts the authors in an unenviable position: They now need to attempt to fix a mistake without any feedback regarding where they (potentially) went wrong.
>
> The goal of the review process is to offer "constructive" feedback and to offer arguments that are "as comprehensive as possible" (quoted from https://iclr.cc/Conferences/2023/ReviewerGuide).  I would argue that this review achieves neither of these goals.  As an author, it can be demoralizing to receive such a review, especially given the dismissive nature of statements such as "it does not reach the requirement of ICLR" when they are accompanied by unsupported arguments.  The so-called "bar" for acceptance is different for each reviewer; and so if the reviewer is going to assert that this paper is below their bar, they should explain exactly what their threshold for acceptance would be.  How could the authors improve the paper?  What would they need to do in order to improve the reviewer's score?
>
> It's essential that we keep in mind that the authors may have spent months working on this paper, and that there may be students involved in the process.  Our goal should be to build a supportive community where we offer constructive feedback and acknowledge the strengths of each submission, rather than a community wherein it is acceptable to dismiss work based on almost no feedback.  Given all of this, I think it would be appropriate for the reviewer to provide additional feedback to the authors.

---

> ### Author Response · Authors · 2022-11-14
> **Response to Reviewer BJCq**
>
> >The novelty is limited to combining the existing methods.
>
> We do not agree. Our contribution is underestimated by the reviewer.
>
> First, the main contribution of this paper is to leverage the topological structure of distributions to address the problem of out-of-distribution (OOD) generalization. We propose a new principled method that seamlessly integrates topological information to distributionally robust optimization (DRO) to develop strong OOD resilience. The topological information can either be acquired from physical topology or learned from data, which enables our approach to be applied to a wide range of tasks including classification, regression, and semantic segmentation. DRO is typically constrained by moment constraints, $f$-divergence, Wasserstein distance or Maximum Mean Discrepancy. However, how to utilize the topology to constrain DRO remains a seldom investigated problem.
> The novelty has been clearly recognized by other reviewers.
> Reviewer uihf commented that  ```"the idea of incorporating the topological prior in this paper is novel and very interesting"```.
> Reviewer vFEm commented that ```"the proposed topology-based DRO is novel enough"```.
> Reviewer bQwH commented that ```"the main idea is novel and of high quality"```.
>
> Second, empirical results in a wide range of tasks including classification, regression, and semantic segmentation demonstrate the superior performance of our method over SOTA (see Tables 1, 2, and 4).
> Specifically, in semantic segmentation, we conducted experiments on Sen1Floods11 [1] which is a public dataset for global flood mapping during extreme weather events. This dataset is closely related to high-stake and safety-critical applications.
> The significance has been recognized by Reviewer uihf: ```"I believe the algorithm in this paper can potentially have some real-world impact on improving the well-being of humans"```.
>
> Third, the explainable distributional topology is consistent with human knowledge and scientific plausibility.
> In our experiments on the Sen1Floods11 dataset, we empirically verified this claim by relating the learned topology to the causes of each flood event. In the learned topology, we found that the most influential events were aroused by the most prevalent disaster such as heavy rainfall, while the least influential events were aroused by edge cases such as dam collapse (see Section 5.3 and Figure 6).
>
> All of these are novel and have not been studied in previous works. We sincerely hope that the reviewer can reconsider the rating.
>
> >The definition of uncertainty set is incorrect.
>
> We disagree with the statement.
> The uncertainty set of our method is defined as an arbitrary mixture of training groups, which is consistent with Group DRO [2] (see Eq. (7)).
>
> >This method is applicable to distributionally robust optimization for graph data.
>
> We think the reviewer has misunderstood the scope of our work.
> Our paper provides a generic solution to OOD generalization for a wide range of data types (synthetic data in Section 5.1, time series data in Section 5.2, and image data in Section 5.3 ), not limited to graph data.
> In the distributional topology, each entity denotes a group rather than a single data point.
>
> Thanks for reviewing our paper. We would appreciate it if you can provide any explanation for the comments "the definition of uncertainty set is incorrect" and "the contributions are all over-claimed".
> We are very happy to respond to any questions or comments during the discussion phase.
>
> [1] Bonafilia, Derrick, et al. "Sen1Floods11: A georeferenced dataset to train and test deep learning flood algorithms for sentinel-1." CVPRW 2020.
>
> [2] Sagawa et al. Distributionally robust neural networks for group shifts: On the importance of regularization for worst-case generalization. ICLR 2019.

---

> > ### Author Response · Authors · 2022-11-23
> > **We would like to hear back from Reviewer BJCq**
> >
> > Dear Reviewer BJCq,
> >
> > We would like to follow up to see if our response addresses your concerns or if you have any further questions.

---

### Official Review · Reviewer_uihf · 2022-10-26

**Confidence:** 4
**Correctness:** 3
**Technical Novelty And Significance:** 3
**Empirical Novelty And Significance:** 3
**Recommendation:** 6

**Clarity, Quality, Novelty And Reproducibility:**

Yes, it's clear, and the algorithm is novel. Code was provided, but I did not run it.

**Strength And Weaknesses:**

Strength:
- I think the idea of incorporating the topological prior in this paper is novel and very interesting. Especially, most of the prior work in dealing with distribution shift will cast the problem as a minimax problem, which is overly pessimistic. Though TRO also adopts such a method, it constrains the search space of maximization with the topological prior. This can (hopefully) exclude those implausible distributions, and further improves the OOD generation performance.

- The experiments conducted in this paper are related to very important real-world problems, e.g., flood prediction. So, I believe the algorithm in this paper can potentially have some real-world impact on improving the well-being of humans.

- TRO also has theoretical guarantees on the convergence rate under both convex and non-convex losses, though the derivations are straightforward from the existing results and the technical contributions are limited.


Weakness/Questions:
- The proposed framework, TRO, seems to only work for data distributions where they can be categorized into a bunch of discrete groups. It's unclear to me whether many problems satisfy this assumption. It would be great if the authors can provide more discussions.

- For those groups that we don't have any data, how can we infer the topological relationship between them with the groups with data? Also, how do we know the number of groups, if it's not given?

- How do you choose $\lambda$ in your experiments? If you use a validation set, how do you construct this validation set; and shouldn't it also suffer from distribution shift?

**Summary Of The Paper:**

This paper proposes an algorithm for improving out-of-distribution generalization performance by modeling the topology structures of distributions. Specifically, the algorithm first constructs a graph based on either manually defined priors, e.g., spacial relationships, or learning it from the data. This topology graph serves as a topological prior. Then, it remains to solve a minimax problem under the constraint induced by the topological prior. In contrast to the conventional DRO approach, the proposed algorithm utilizes the topological structure and hence reduces the search space. This can exclude those implausible distributions, reduce the pessimism and hence improve the OOD generalization. The authors provide theoretical guarantees for the convergence rate of the algorithm under both convex and nonconvex loss functions. In addition, the experimental results demonstrate the effectiveness of the algorithm.


**Summary Of The Review:**

This paper propose a novel algorithm for utilizing the topological structure to improve the OOD generalization performance. I don't have critical concerns about this paper, except for some minor questions/weakness. Hence, I would recommend for an acceptance.

---

> ### Author Response · Authors · 2022-11-14
> **Response to Reviewer uihf**
>
> Thank you for your encouraging comments. We are glad that you find our idea is ```"novel and very interesting"```, our experiments ```"are related to very important real-world problems"```, and ```"believe the algorithm can potentially have some real-world impact on improving the well-being of humans"```. Below we address your questions one by one.
>
> **Q1. More discussion on the number of discrete groups.**
>
> **Number of discrete groups.** In our experiments on DG-60 [1] dataset, we only use 6 groups (10\%) as the training set and use the other 54 groups (90\%) as the test set.
> In Table 1, we empirically show that TRO outperforms ERM and Group DRO by 14.7\% and 11.13\%, which demonstrates the effectiveness of TRO with a small number of training groups.
>
> **Whether many problems satisfy this assumption.** In common benchmarks for Out-of-Distribution generalization such as DomainBed [2] and WILDS [3], most datasets consist of a bunch of discrete groups.
> For example, in WILDS, 8 out of 10 datasets have more than 40 discrete groups (see Fig. 2 in [3]).
>
> **Q2. For those groups that we don't have any data, how can we infer the topological relationship between them with the groups with data? Also, how do we know the number of groups, if it's not given?**
>
> For data-driven topology, since the data of test groups are inaccessible, we can only infer the topological relationship within training groups. The underlying assumption is that training groups with high centrality also exert a strong influence on test groups.
> We visualize the data-driven topology and group centrality on TPT-48 [4] and Sen1Floods11 [5] datasets in Figs. 4 and 6.
> Specifically, we empirically show the learned topology enjoys strong explainability (Sec. 5.3 and Fig. 6).
> In Tables 1, 2, and 4, we empirically show that data-driven topology consistently yields better performance than physical topology, indicating the data topology may capture the distributional distances more accurately.
> We summarize the results below:
>
> |              | DG-15&uarr; | DG-60&uarr; | TPT-48&darr; | Sen1Floods11&uarr; |
> |--------------|:-----:|:-----:|:------:|:------------:|
> | ERM          | 58.00 | 76.02 |  1.426 |     .430     |
> | Group DRO    | 43.22 | 79.59 |  1.356 |     .433     |
> | TRO (physic) | 67.56 | 89.19 |  1.177 |       -      |
> | TRO (data)   | **67.89** | **90.72** |  **1.129** |     **.450**   |
>
> If the number of training groups is not given, a.k.a unknown group partitions, we can leverage existing methods such as [6] to infer the latent group partitions. After that, we can leverage topology learning to measure the connectivity between groups.
>
> **Q3. More details and discussion on the validation set.**
>
> **How to construct this validation set?** We choose $\lambda$ using the validation dataset. As stated in Section 5, following [2], we perform model selection based on a validation set constructed from training groups only.
> In detail, we split each training group into training (80\%) and validation (20\%) subsets. Then, we pool the validation subsets of each training group to create an overall validation set.
> As shown in Figure 7, we reported both mean and standard deviation on the validation set of the Sen1Floods11 dataset across different $\lambda$.
>
> **Shouldn't it also suffer from distribution shift?** You are right. Such validation set also suffers from the distributional shift.
> How to construct a validation set to select promising parameters for unseen test distributions is still an open question.
> As shown in [2], typically, there are two popular model selection strategies for OOD generalization: training group validation set and leave-one-group-out cross-validation. Our model selection strategy belongs to the former category.
> [2] empirically show that neither of these two approaches can select the optimal parameters for test groups, however, model selection with a training group validation set outperforms leave-one-group-out cross-validation across multiple datasets and algorithms (see Section 5.1 in [2]).
>
> Please let us know if our response addresses your concerns or if you have any further questions. We would really appreciate the opportunity to discuss this further if our response has not already addressed your concerns. Thank you again!
>
> [1] Xu, Zihao, et al. "Graph-Relational Domain Adaptation." ICLR 2022.
>
> [2] Gulrajani, Ishaan, and David Lopez-Paz. "In search of lost domain generalization." ICLR 2021.
>
> [3] Koh et al. "Wilds: A benchmark of in-the-wild distribution shifts." ICML 2021.
>
> [4] Vose, R., et al. "Gridded 5km GHCN-daily temperature and precipitation dataset (nCLIMGRID) version 1." Maximum Temperature, Minimum Temperature, Average Temperature, and Precipitation 2014.
>
> [5] Bonafilia, Derrick, et al. "Sen1Floods11: A georeferenced dataset to train and test deep learning flood algorithms for sentinel-1." CVPRW 2020.
>
> [6] Creager, Elliot, Jörn-Henrik Jacobsen, and Richard Zemel. "Environment inference for invariant learning."  ICML 2021.

---

> ### Author Response · Authors · 2022-11-18
> **Strong and explainable results on standard benchmark DomainBed**
>
> Dear Reviewer uihf,
>
> As suggested by Reviewer bQwH, we conducted experiments on the standard benchmark DomainBed [1]. Following the instructions of the official implementation of DomainBed [1], we have conducted experiments on PACS [2], Terra [3], and VLCS [4]. Image samples of the three datasets are shown in Figure 12 (left) of Appendix 8.3.
>
> (1) PACS is one of the most popular datasets for out-of-distribution generalization. It consists of images from four groups: "Art", "Cartoon", "Photo" and "Sketch".
> Results on PACS are shown below. "Art": "Art" is the test group while the other three groups are training groups.
>
> |              |    Art    |  Cartoon  |   Photo   |   Sketch  | Average   |
> |--------------|:---------:|:---------:|:---------:|:---------:|-----------|
> | ERM          | **88.1**(0.1) | 77.9(1.3) |  97.8(0)  | 79.1(0.9) | 85.7(0.5) |
> | Group DRO    | 86.4(0.3) | 79.9(0.8) | **98.0**(0.3) | 72.1(0.7) | 84.1(0.4) |
> | CORAL (SOTA) | 87.7(0.6) | 79.2(1.1) | 97.6(0)   | **79.4**(0.7) | 86.0(0.2) |
> | TRO (ours)   | 87.7(0.5) | **82.1**(0.5) | **98.0**(0.2) | 78.2(1.9) | **86.5**(0.4) |
>
> As observed, in average accuracy, TRO outperforms the SOTA by 0.5\%.
> To further investigate the results, we visualize the learned topology in Figure 12 (right) of Appendix 8.3.
> As observed, when "Cartoon" is the test group, the topology is a chain graph consisting of three nodes where "Art" is the most influential group. We assume the reason is "Art" may contain more information than  "Photo" and "Sketch" as "Art" is the combination of photos and various kinds of styles.
> **Even though the topology is so simple, it enables our method to significantly outperforms ERM and DRO by 0.8\% and 2.4\% in average.** The results empirically demonstrate the strong effectiveness and explainability of our method when the number of training groups is quite limited, i.e., 3.
>
> We would like to point out that when the distributional shift across different groups is small (see explanation on the results of VLCS), the influential group may not exist and all groups share the same centrality. In this special case, TRO aims to strike a good balance between the average (ERM) risk and the worst-case (DRO) risk.
>
> (2) Terra consists of images of wild animals captured by camera traps under four locations.
> Results on Terra are shown below.
>
> |              |    L100    |  L38  |   L43   |   L46  | Average   |
> |--------------|:---------:|:---------:|:---------:|:---------:|-----------|
> | ERM          | 50.8(1.8) | 42.5(0.7) |  57.9(0.6)  | 37.6(1.2) | 47.2(0.4) |
> | Group DRO    | 47.2(1.6) | 40.1(1.6) | 57.6(0.9) | **43.0**(0.7) | 47.0(0.3) |
> | MMD (SOTA) | 52.2(5.8) | **47.0**(0.6) | 57.8(1.3)   | 40.3(0.5) | **49.3**(1.4) |
> | TRO (ours)   | **53.2**(2.4) | 42.2(1.3) | **59.0**(0.8) | 41.3(0.5) | 49.0(0.6) |
>
> As observed, in average accuracy, TRO achieves comparable results with the SOTA, and outperforms ERM and DRO by 1.8\% and 2.0\%.
>
> (3) VLCS is a combination of four datasets: Caltech101, LabelMe, SUN09, and VOC2007. It consists of photos of five classes: "bird", "car", "chair", "dog", and "person". Results on VLCS are shown below.
>
> |              |    Caltech101    |  LabelMe  |   SUN09   |   VOC2007  | Average   |
> |--------------|:---------:|:---------:|:---------:|:---------:|-----------|
> | ERM          | 97.6(1.0) | 63.3(0.9) |  72.2(0.5)  | 76.4(1.5) | 77.4(0.3) |
> | Group DRO    | 97.7(0.4) | 62.5(1.1) | 70.1(0.7) | **78.4**(0.9) | 77.2(0.6) |
> | DANN (SOTA) | **98.5**(0.2) | 64.9(1.1) | **73.1**(0.7)   | 78.3(0.3) | **78.7**(0.3) |
> | TRO (ours)   | 96.9(0.2) | **65.0**(0.8) | 71.3(0.9) | 75.5(0.9) | 77.2(0.5) |
>
> As observed, the average accuracy of DRO and TRO is the same.
> We assume the reason is that the distributional shift across different groups is small [2], and therefore the influential group may not exist and all groups share the same centrality.
> In this case, TRO aims to strike a good balance between the average (ERM) risk and the worst-case (DRO) risk.
> The images of VLCS are all photos and the distributional shift is not as significant as PACS (e.g., Photo vs. Sketch).
> As stated in Section 2.1 of [2], "Despite the famous analysis of dataset bias that motivated the creation of the VLCS benchmark, it was later shown that the domain shift is much smaller with recent deep features", and PACS [2] was proposed to address this limitation.
>
> Thank you for your valuable comments. Please let us know if you have any suggestions that can be included in the revised paper.
>
> [1] Gulrajani, Ishaan, and David Lopez-Paz. "In search of lost domain generalization." ICLR 2021.
>
> [2] Li et al. "Deeper, broader and artier domain generalization." ICCV 2017
>
> [3] Beery et al. "Recognition in terra incognita." ECCV 2018
>
> [4] Fang et al. "Unbiased metric learning: On the utilization of multiple datasets and web images for softening bias." ICCV 2013

---

### Author Response · Authors · 2022-11-21
**General response**

Dear Reviewers, Area Chairs, and Program Chairs,

Thank you for your time and effort in reviewing our paper. We appreciate the reviewers find our idea ``"novel"`` (uihf, vFEm, bQwH), ``"very interesting"``(uihf), and ``"of high quality"``(bQwH), our method ``"theoretically guaranteed"``(uihf, vFEm), the experiments ``"pretty sufficient"``(vFEm) and ``"related to very important real-world problems, e.g., flood prediction"``(uihf), the results ``"strong"``(bQwH) and show  ``"effectiveness"`` (vFEm). We also appreciate they find the writing is ``"clear"``(uihf), ``"easy to follow"``(BJCq), and ``"solid"``(bQwH).
We have responded to the individual comments of each reviewer and carefully revised the paper. All revisions are highlighted in light blue color.

**Contribution of this work.**  The main contribution of this paper is to leverage the topological structure of distributions to address the problem of out-of-distribution (OOD) generalization. We propose a new principled method that seamlessly integrates topological information to distributionally robust optimization (DRO) to develop strong OOD resilience. DRO is typically constrained by moment constraints, $f$-divergence, Wasserstein distance or Maximum Mean Discrepancy. However, how to utilize the topology to constrain DRO remains a seldom investigated problem.

**Summary of reviews.**
Reviewers uihf and vFEm vote for acceptance. Reviewers BJCq, zKvZ, and bQwH vote for reject.
We summarize the comments of the three reviewers below:

1. The major concern of Reviewer BJCq is the novelty of this work. However, the novelty has been clearly recognized by other reviewers: ``"the idea of incorporating the topological prior in this paper is novel and very interesting"``(uihf), ``"the proposed topology-based DRO is novel enough"``(vFEm), and ``"the main idea is novel and of high quality"``(bQwH). Moreover, the reviewer did not provide any explanation for the comments such as "the definition of uncertainty set is incorrect" and "the contributions are all over-claimed".
Without any further explanation, it is hard for us to respond to these comments, especially when the reviewer may have potential misunderstandings of our work (e.g., the reviewer thinks our method is limited to graph data).
If the reviewer had provided any explanation for these comments, we could have probably addressed the concerns.

2. Reviewer zKvZ did not provide any comment and the confidence is 1. Hence, this is not a valid review.

3. The major concerns of Reviewer bQwH are the reproducibility and the lack of experiments on standard benchmarks.
To address the issue of reproducibility, we have included sufficient details on both the method and experiment in the revised paper, and also provided source codes in the supplementary. The reproducibility has been recognized by other reviewers: ``"the study reported in sufficient detail to allow for its reproducibility"``(BJCq), and ``"implementation details and codes are provided to ensure reproducibility"``(vFEm).
To address the issue of standard benchmarks, we have conducted experiments on DomainBed.
We reported the experimental results on three datasets PACS, Terra, and VLCS (see Tabs. 5-7). We also visualized the learned topology (see Fig. 12) and provided further explanation. **Even though the topology (a chain graph of three nodes) is so simple, it enables our method to significantly outperforms ERM and DRO by 0.8% and 2.4% on PACS**. We also uncover a special case of our method (see explanation on VLCS): when the distributional shift across different groups is small, all groups may share the same centrality and the influential group may not exist. In this case, our method aims to strike a good balance between the average (ERM) risk and the worst-case (DRO) risk.
Despite the promising results on DomainBed, we argue that the experiments in the main body are sufficient to demonstrate the effectiveness of our method and the lack of experiments on standard benchmarks is not a drawback.
First, we use the same model selection strategy as DomainBed did. All baselines and their implementations are also from DomainBed.
Second, we conducted experiments on Sen1Floods11, a public dataset for global flood mapping and closely related to high-stake applications. The significance has been recognized by Reviewer uihf: ``"I believe the algorithm in this paper can potentially have some real-world impact on improving the well-being of humans"``.
Third, DomainBed only has a single task: image classification, while our experiments cover classification, regression, and semantic segmentation.
Additionally, we have done our best to eliminate all concerns of the reviewer by providing point-to-point responses.

We believe that these clarifications and additional results have improved the paper, and kindly ask the reviewers to take this into account when considering score adjustments. We welcome any further discussion with the reviewers.

---

### Decision · Program_Chairs · 2023-01-20

**Decision:**

Accept: poster

**Justification For Why Not Higher Score:**

Because the improvement in Flood prediction was not significant and the proposed approach was not tested on a wide variety of datasets with geographical meta-data.

**Justification For Why Not Lower Score:**

Because the proposed approach to take topological information in into account is novel and improves performance.

**Metareview: Summary, Strengths And Weaknesses:**

This paper suggests taking advantage of topological information about data source into account using a graph and proposes a method that relies on such a graph for a more robust prediction.

The paper is well-written and easy to follow. The main idea is novel and interesting. Authors show that the proposed algorithm leads to improvement in both synthetic and realistic settings. Therefore, I recommend acceptance.

**Note From Pc:**

if the above contains the word "oral" or "spotlight" please see: "oral" presentation means -> notable-top-5% and "spotlight" means -> notable-top-25%. As stated in our emails, we are disassociating presentation type from AC recommendations

**Summary Of Ac-Reviewer Meeting:**

The main contributions of the paper was discussed.

Reviewer bQwH: The idea of using graphs for describing relationship between distributions is interesting.

Reviewr uihf: The problem formulation is interesting. The results on flood prediction is promising and interesting.

Both reviewers thought the paper should be accepted. Reviewers zKvZ and BJCq who were favoring rejection did not participate in the meeting. In particular, there was a consensus among AC and reviewers that the review of the reviewer BJCq has very low quality and substance and that should be taken into account when making the final decision.